# GCVR: Reconstruction from Cross-View Enable Sufficient and Robust Graph Contrastive Learning

**Qianlong Wen**[1]   **Zhongyu Ouyang**[1]   **Chunhui Zhang**[2]   **Yiyue Qian**[1]   **Chuxu Zhang**[3]   **Yanfang Ye**[1]

[1]University of Notre Dame, Notre Dame, IN, USA
[2]Dartmouth College, Hanover, NH, USA
[3]Brandeis University, Waltham, MA, USA

## Abstract

Among the existing self-supervised learning (SSL) methods for graphs, graph contrastive learning (GCL) frameworks usually automatically generate supervision by transforming the same graph into different views through graph augmentation operations. The computation-efficient augmentation techniques enable the prevalent usage of GCL to alleviate the supervision shortage issue. Despite the remarkable performance of those GCL methods, the InfoMax principle used to guide the optimization of GCL has been proven to be insufficient to avoid redundant information without losing important features. In light of this, we introduce the **G**raph **C**ontrastive Learning with Cross-**V**iew **R**econstruction (GCVR), aiming to learn robust and sufficient representation from graph data. Specifically, GCVR introduces a cross-view reconstruction mechanism based on conventional graph contrastive learning to elicit those essential features from raw graphs. Besides, we introduce an extra adversarial view perturbed from the original view in the contrastive loss to pursue the intactness of the graph semantics and strengthen the representation robustness. We empirically demonstrate that our proposed model outperforms the state-of-the-art baselines on graph classification tasks over multiple benchmark datasets.

## 1 INTRODUCTION

As a kind of ubiquitous data form, graph-structured data is known for modeling complex interaction systems in the real world. Among the existing techniques proposed to model the patterns behind graph-structured data, Graph Neural Networks (GNNs) [Kipf and Welling, 2017, Veličković et al., 2018, Hamilton et al., 2017, Xu et al., 2019] have

achieved remarkable performance and thereby been employed in many applications, like user preference prediction, recommendation systems, and molecule property prediction [McAuley et al., 2015, Hu et al., 2020a, Wen et al., 2023, Ouyang et al., 2024]. Despite their success, GNNs are often constrained by the supervised learning paradigm, which necessitates a substantial volume of labeled data—a requirement that is frequently costly or impractical. Therefore, extracting the rich underlying knowledge from the unlabeled graphs has attracted increasing attention and stimulated a series of research about self-supervised learning (SSL) on graphs [Qiu et al., 2020, Hassani and Khasahmadi, 2020, Sun et al., 2019, Zhao et al., 2021a], where only minimal or no labels are required. Existing graph SSL strategies will design different pre-training tasks for optimization to fully utilize the information from unlabeled graphs, where graph contrastive learning (GCL) follows the mutual information maximization principle (InfoMax) [Veličković et al., 2019] to maximize the agreements of the positive pairs while minimizing that of the negative pairs in the embedding space. However, the GCL paradigm has been empirically and theoretically proved to be insufficient to learn robust and transferable representation [Suresh et al., 2021, Zhao et al., 2021b]. State-of-the-art GCL methods [Qiu et al., 2020, Hassani and Khasahmadi, 2020, You et al., 2020] usually rely on applying specific graph augmentation operations (e.g., Edge Removing and Subgraph Sampling) on anchor graph $G$ to generate a positive pair of samples. Then, the graph feature encoder $f$ will be trained to ensure representation consistency within the positive pair. Thus, the choice of augmentation operators and their strength can yield significant impacts on the final optimization result. Moderate graph augmentation will push encoders to capture redundant and biased information [Tschannen et al., 2019], adversely impacting the transferability of representations through "shortcut" solutions [Geirhos et al., 2020, Minderer et al., 2020, Robinson et al., 2021]. This is visually depicted in Figure 1(a), where the shared part of the two augmentation views includes both predictive information (the overlapping area with $y$) and non-predictive informa-

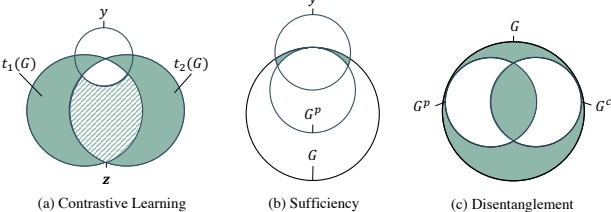

(a) Contrastive Learning     (b) Sufficiency     (c) Disentanglement

Figure 1: Illustration of the relation between graph $G$, label $y$, predictive feature subsets $G^p$ and non-predictive feature subset $G^c$ in terms of information entropy. Ideally, the green areas in the three figures are null. (a) The usual optimization result of graph contrastive learning, where the shared features of two augmentation views are extracted for the learned representation $\mathbf{z}$. Owing to the lack of supervision or domain knowledge, redundant and biased information (shadow area) is usually included in $\mathbf{z}$; (b) $G^p$ cover the feature subset which is sufficient to make correct graph label identification ($I(y; G \mid G^p) = 0$), other features ($G^c$) is either useless or misguiding; (c) $G^p$ and $G^c$ are supposed to be mutually excluded with each other ($I(G^p; G^c) = 0$). The union of them covers all the features of the original data.

tion (shadow area). This phenomenon is also empirically proved by many previous GCL works [Suresh et al., 2021, Li et al., 2022, Yang et al., 2021], where lower contrastive loss does not necessarily lead to better performance and robustness, especially under the out-of-distribution (OOD) setting [Ye et al., 2021]. Conversely, overly aggressive augmentation also proves suboptimal, as it indiscriminately disregards both predictive and trivial features in the absence of additional guiding knowledge [Chen et al., 2021].

To address the dilemma, recent works [Suresh et al., 2021, Li et al., 2022, You et al., 2021] propose to modify the existing graph augmentation techniques in an automated manner. These methods operate under the premise that the most salient substructures, which are resistant to graph augmentation, are adequate for downstream label identification. Consequently, they introduce trainable regularization on the graph topological structure to eliminate trivial substructures. While these approaches mitigate the feature suppression issue to a degree, they are constrained by the same optimization principles, inheriting similar limitations. The harsh regularization on graph topology tends to narrow the focus of encoders to 'shallower' features, such as graph size and central node, in the absence of external knowledge [Bevilacqua et al., 2021], thereby impairing generalization in tasks requiring more comprehensive understanding. Therefore, the GCL methods, guided by the saliency philosophy, have not yet optimally balanced representation sufficiency and robustness. To learn better graph representation, an SSL graph method that can better reconcile the information redundancy and sufficiency is in urgent need. The optimal representation, as suggested by the information bottleneck

(IB) principle [Tishby et al., 2000], should extract minimal yet sufficient information during the learning process. Empirical evidence supports the superiority of representations aligning with the IB principle, demonstrating enhanced robustness and transferability across various domains [Wu et al., 2020]. Consider an input graph $G$, with $G^p$ representing its predictive feature subset and $G^c$ the complementary non-predictive feature subset. Recent studies on rationale invariance discovery [Wu et al., 2022b,a] suggest that these subsets satisfy the sufficiency condition $I(y; G \mid G^p) = 0$ and the disentanglement condition $I(G^p; G^c) = 0$. The relationships among $G$, $G^p$, and $G^c$ are depicted in Figure 1 (b) and (c). The representation ideally adhering to the IB principle would include all the features in $G^p$ while minimizing the information in $G^c$. Although it is impossible to eliminate the redundant information across different downstream tasks since there will be a variance between the knowledge required for different applications, a graph representation less suppressed by $G^c$ is expected to generalize better on different downstream tasks. Furthermore, achieving this goal in the self-supervised setting is even more challenging.

To fill this gap, we propose **G**raph Contrastive Learning with **C**ross-**V**iew **R**econstruction (GCVR), to pursue the target optimization objective. GCVR is comprised of a graph encoder and two distinct decoders, each specialized in extracting information pertinent to predictive and non-predictive features, respectively. The model endeavors to fulfill the disentanglement objective through an innovative cross-view representation reconstruction scheme. This scheme involves both intra-view and inter-view reconstructions, aiming to reconstruct the initial learned representation using the bifurcated feature subsets. Furthermore, the encoded representation from the original view perturbed in the adversarial fashion serves as the third view when computing the contrastive loss, apart from the predictive relevant representations of the two augmentation views, to further improve the representations' robustness and prevent them from collapsing into partial solution. We present a theoretical analysis illustrating GCVR's proficiency in approximating the Information Bottleneck (IB) principle, thus enhancing representation robustness without compromising on sufficiency. Finally, empirical validation of GCVR's effectiveness is conducted through extensive experiments on public graph benchmark datasets. The experimental results demonstrate that GCVR achieves significant performance gains over different datasets and settings compared with state-of-the-art baselines. The main contributions of this work are summarized from three aspects: (i) We propose the GCVR to alleviate the feature suppression issue with the cross-view reconstruction mechanism; (ii) We provide solid theoretical analysis on our model designs; (iii) Thorough experiments are conducted to demonstrate the robustness and transferability of the learned representations via GCVR. The source code of our proposed GCVR is publicly available at https://github.com/HoytWen/GCVR

## 2 PRELIMINARIES

### 2.1 GRAPH REPRESENTATION LEARNING

In this work, we focus on the graph-level task, let $\mathcal{G} = \{G_i = (V_i, E_i)\}_{i=1}^{N}$ denote a graph dataset with N graphs, where $V_i$ and $E_i$ are the node set and edge set of graph $G_i$, respectively. We use $x_v \in \mathbb{R}^d$ and $x_e \in \mathbb{R}^d$ to denote the attribute vector of each node $v \in V_i$ and edge $e \in E_i$. Each graph is associated with a label, denoted as $y_i$, the goal of the graph representation learning is to learn an encoder $f : G_i \to \mathbb{R}^d$ so that the learned representation $\mathbf{z}_i = f(G_i)$ is sufficient to predict $y_i$ related to the downstream task. We clarify the sufficiency of $\mathbf{z}_i$ as containing no less information of the label of $G_i$ [Achille and Soatto, 2018], and it is formulated as:

$$I(G_i; y_i \mid \mathbf{z}_i) = 0, \tag{1}$$

where $I(;)$ denotes the mutual information between two variables.

### 2.2 CONTRASTIVE LEARNING

Contrastive Learning (CL) is a self-supervised representation learning method that leverages instance-level identity for supervision. During the training phase, each graph $G$ firstly goes through proper data augmentation to generate two data augmentation views $t_1(G)$ and $t_2(G)$, where $t_1(\cdot)$ and $t_2(\cdot)$ are two augmentation operators. Then, the CL method encourages the encoder $f$ (a backbone network plus a projection layer) to map $t_1(x)$ and $t_2(x)$ closer in the hidden space so that the learned representations $\mathbf{z}_1$ and $\mathbf{z}_2$ maintain all the information shared by $t_1(G)$ and $t_2(G)$. The learning of the encoder is usually directed by a contrastive loss, such as NCE loss [Wu et al., 2018b], InfoNCE loss [van den Oord et al., 2018], and NT-Xent loss [Chen et al., 2020]. In Graph Contrastive Learning (GCL), we usually adopt a GNN, such as GCN [Kipf and Welling, 2017] or GIN [Xu et al., 2019], as the backbone network, and the commonly-used graph data augmentation operators [You et al., 2020], such as node dropping, edge perturbation, subgraph sampling, and attribute masking.

All the GCL-based methods are built on the assumption that augmentations do not break the sufficiency requirement to make correct predictions. Here, we follow [Federici et al., 2020] to clear up the definition of mutual redundancy. $t_1(G)$ is redundant to $t_2(G)$ with respect of $y$ iff $t_1(G)$ and $t_2(G)$ share the same predictive information. Mathematically, the mutual redundancy in CL exists when:

$$I(t_1(G); y \mid t_2(G)) = I(t_2(G); y \mid t_1(G)) = 0. \tag{2}$$

Although GCL-based methods are usually capable of extracting useful information for label identification, it is unavoidable to include non-predictive features under the SSL setting owing lack of explicit domain knowledge. There

exists the situation (e.g., OOD setting) that the latent space of learned representation is dominated by non-predictive features in SSL [Chen et al., 2021] and is not informative enough to make the correct prediction. Therefore, feature suppression is not just a prevalent issue in supervised learning, but also in SSL.

### 2.3 FEATURE SUPPRESSION

In this section, we will follow the previous works [Chen et al., 2021, Robinson et al., 2021] to present a more formal definition of feature suppression and clarify its relation with contrastive learning. First of all, we assume graph data $G$ has $n$ feature sub-spaces, $G^1, \ldots, G^n$, where each $G^i \in G$ corresponds to a distinct feature of $G$. To quantify the relation between $G$ and its feature sub-spaces, we need to measure the conditional probability of $G$ given a specific kind of feature sub-space $G^i$ ($i \subseteq [n]$), denoted as $p(G \mid G^i)$. Finally, we define an injective map $g : G^i \to G$ to produce observation $G = g(G^i)$. Due to the reason that $G^i$ is not explicit, so we aim to train an encoder $f : G_i \to \mathbb{R}^d$ to map input graph data $G$ into a latent space to extract useful high-level information $\mathbf{z}^i$ corresponding to each feature sub-space $G^i$ of input data $G$ during contrastive learning. Therefore, we use $p(G \mid \mathbf{z}^i)$ as the approximated value of the measurement $p(G \mid G^i)$. Then we have,

- For any feature sub-space $G^i$ and its complementary feature sub-subspace $G^{\bar{i}}$, $f$ suppress feature $i \subseteq [n]$ if we have $p(G \mid \mathbf{z}^i) = p(G \mid \mathbf{z}^{\bar{i}})$

- For any feature sub-space $G^i$ and its complementary feature sub-subspace $G^{\bar{i}}$, $f$ distinguish feature $i \subseteq [n]$ if $p(G \mid \mathbf{z}^i)$ and $p(G \mid \mathbf{z}^{\bar{i}})$ have disjoint support.

To sum up, a feature is suppressed if it does not make any difference to the instance discrimination. One of the common acknowledgments for unsupervised learning strategy is that it can usually produce representation with uniform feature space distribution due to the lack of supervision, i.e., every feature sub-space is equally treated without feature suppression. However, it could not be the situation in contrastive learning. Taking the commonly used InfoNCE [van den Oord et al., 2018] as an example, it can be divided into two parts, i.e. align term and uniform term [Chen et al., 2020], as follows:

$$\tau \mathcal{L}^{\text{InfoNCE}} = \underbrace{-\frac{1}{m} \sum_{i,j} \text{sim}\left(\boldsymbol{z}_i, \boldsymbol{z}_j\right)}_{\mathcal{L}_{\text{alignment}}}$$
$$+ \underbrace{\frac{\tau}{m} \sum_i \log \sum_{k=1}^{2m} \mathbf{1}_{[k \neq i]} \exp\left(\text{sim}\left(\boldsymbol{z}_i, \boldsymbol{z}_k\right)/\tau\right)}_{\mathcal{L}_{\text{uniform}}}. \tag{3}$$

Aligning the positive pair will distinguish the shared feature

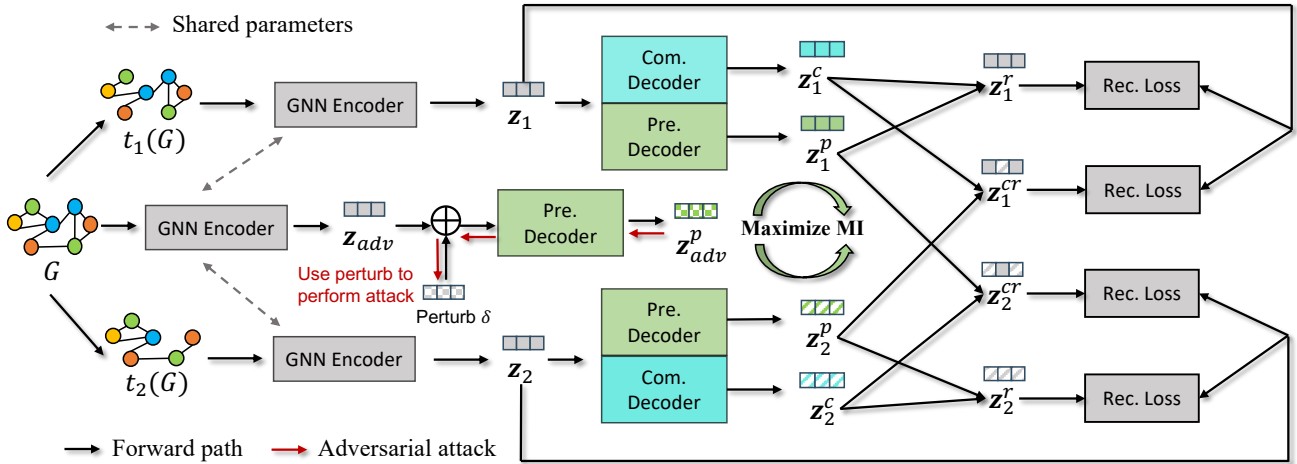

Figure 2: The illustration of the proposed GCVR. (1) Graph augmentations are applied to the input graph $G$ to produce two augmented graphs, which are then fed into the shared graph encoder $f(\cdot)$ to generate two graph embeddings $\mathbf{z}_1$ and $\mathbf{z}_2$. (2) $\mathbf{z}_1$ and $\mathbf{z}_2$ are used as the inputs of the two decoders to generate two pairs of graph embeddings, $\mathbf{z}^p$ captures the predictive factors and $\mathbf{z}^c$ keep other complementary non-predictive features. Then we use the two pairs of representations to reconstruct $\mathbf{z}_1$ and $\mathbf{z}_2$ in both the intra-view and inter-view. (3) An adversarial sample generated by $G$ goes through the same procedure to generate $\mathbf{z}_{adv}^p$. We take it as the third view besides $\mathbf{z}_1^p$ and $\mathbf{z}_2^p$ in CL guarantee the $\mathbf{z}^p$ can guarantee the robustness.

subspace $G^i$. Meanwhile, there also exits random negative samples that might own the same factors in $G^i$, so the uniform term might suppress the feature sub-space $G^i$. Therefore, for any feature $i \subseteq [n]$, the optimization process can either suppress or distinguish it, but both of them can reach lower contrastive loss. From the analysis, we can derive the conclusion mentioned in Section 1 that lower contrastive loss might not yield better performance.

## 3  PROPOSED MODEL

In this section, we introduce the details of our proposed GCVR whose framework is shown in Figure 2. Corresponding theoretical analyses are provided to justify the rationality of our designs. Before diving into the details of GCVR, we briefly introduce the overall framework of our model. Given $G$ as the input graph instance and $f(\cdot)$ as the graph encoder, we add two decoders to map the graph representation $\mathbf{z} = f(G) \in \mathbb{R}^d$ into two different feature spaces $(\mathbf{z}^p, \mathbf{z}^c)$, where $\mathbf{z}^p \in \mathbb{R}^d$ is expected to be specific to the predictive information $G^p$, and $\mathbf{z}^c \in \mathbb{R}^d$ is optimized to elicit the complementary non-predictive factors $G^c$. Later, we reconstruct the representation $\mathbf{z}$ with the feature subsets mapped from the same and different augmentation views to reduce the overlapping between $\mathbf{z}^p$ and $\mathbf{z}^c$. By doing so, we approximate the disentanglement objective demonstrated in Figure 1 and the $\mathbf{z}^p$ is optimized to be the invariant part across different augmentation views. More importantly, the proposed reconstruction procedure and added adversarial views will push the learned $\mathbf{z}^p$ to be as comprehensive and robust as possible for the convenience of representation re-

construction. To sum up, instead of implementing harsh regularization on the graph structure, our GCVR proposed a new optimization strategy to elicit the most predictive features with the reconstruction task, thereby alleviating the feature suppression issue of the cost of information sufficiency. More details of GCVR will be introduced next.

### 3.1  DISENTANGLEMENT BY CROSS-VIEW RECONSTRUCTION

In GCL, we usually leverage a graph encoder, such as a GCN [Kipf and Welling, 2017] or a GIN [Xu et al., 2019], to encode the graph data into its representation. There are multiple choices of graph encoders in GCL, including GCN [Kipf and Welling, 2017] and GIN [Xu et al., 2019], etc. In this work, we adopt GIN as the backbone network $f$ for simplicity. Note that any other commonly used graph encoders can also be adapted to our model. Given two augmentation views $t_1(G)$ and $t_1(G)$ (where $t_1(\cdot)$ and $t_2(\cdot)$ are IID sampled from the same family of augmentation $\mathcal{T}$), we firstly use the encoder $f(\cdot)$ to map them into a lower dimension hidden space for the two embeddings, $\mathbf{z}_1$ and $\mathbf{z}_2$. Instead of directly maximizing the agreement between the two representations $\mathbf{z}_1$ and $\mathbf{z}_2$, we further feed each of them into a pair decoders $(g_p, g_c)$ (both of them are MLP-based networks or GNN and they share the same architecture) and optimize the two decoders to map each of the presentations into the two disentangled feature sub-spaces:

$$[\mathbf{z}^p = g_p(f(t(G))), \ \mathbf{z}^c = g_c(f(t(G)))], \quad (4)$$

where a pair of embeddings for both $t_1(G)$ and $t_2(G)$ are generated. Ideally, $\mathbf{z}_1^p$ and $\mathbf{z}_2^p$ suffice the mutual redundancy assumption stated in 2.2 because $t_1(G)$ and $t_1(G)$ are augmented from the same original graph, and thus naturally share the same predictive factors.

Here, we clarify the lower bound of the mutual information between the augmented view $t_1(G)$ and the two learned predictive representations in Theorem 1 and we can get the same conclusion with another augmented view $t_s(G)$.

**Theorem 1.** *Suppose $f(\cdot)$ is a GNN encoder as powerful as 1-WL test. Let $\mathbf{z}_1^p$ and $\mathbf{z}_2^p$ be specific to the predictive information of $G$, meanwhile $\mathbf{z}_1^c$ and $\mathbf{z}_2^c$ account for the complementary non-predictive factors of $t_1(G)$ and $t_2(G)$. Then we have:*

$$I\left(t_1(G); \mathbf{z}_2^p, \mathbf{z}_2^c\right) \geqslant I\left(\mathbf{z}_1^p; \mathbf{z}_2^p\right),$$

where $G \in \mathcal{G}$ and $t_1(\cdot), t_2(\cdot) \in \mathcal{T}$. The detailed proof is provided in Appendix E. Given the lower bound, we substitute the objective by the mutual information between the two representations in the predictive view ($\mathbf{z}_1^p$ and $\mathbf{z}_2^p$) to maximize the consistency between the information of the two views. Therefore, we derive the objective function ensuring view invariance as follows:

$$\mathcal{L}_{\text{pre}} = \frac{1}{N} \sum_{i=1}^{N} \mathcal{L}_{\text{CL}}(\mathbf{z}_{1,i}^p, \mathbf{z}_{2,i}^p), \qquad (5)$$

where $\mathcal{L}_{\text{CL}}(\cdot)$ is the adopted InfoNCE loss [van den Oord et al., 2018]. To further pursue the feature disentanglement as illustrated in Figure 1(c), we propose the cross-view reconstruction mechanism. To be specific, we would like the representation pair ($\mathbf{z}^p, \mathbf{z}^c$) within and cross the augmentation views to be able to recover the raw data so that the two objectives can be approached simultaneously. Due to the fact that graphs are non-Euclidean structured data, we instead try to recover $\mathbf{z} = f(t(G))$ given ($\mathbf{z}^c$ and $\mathbf{z}^p$).

More specifically, we first perform the reconstruction within the augmentation view, namely mapping ($\mathbf{z}_w^p, \mathbf{z}_w^c$) to $\mathbf{z}_w$, where $w \in \{1, 2\}$ representing the augmentation view. Then, we define the ($\mathbf{z}_{w'}^p, \mathbf{z}_w^c$) as a cross-view representation pair and the reconstruction procedure is repeated on it to predict $\mathbf{z}_w$, aiming to minimize the overlapping between $\mathbf{z}^p$ and $\mathbf{z}^c$, where $w = 1, w' = 2$ or $w = 2, w' = 1$. Intuitively, the reconstruction process is capable of separating the information of the shared feature sets from the one resided in the unique feature sets between the two augmentation views. Since the two IID sampled augmentation operators ($t_1(\cdot)$ and $t_2(\cdot)$) are expected to preserve the predictive/rational features while varying the augmentation-related ones, we disentangle the rational features from $G$ according to the rationale discover studies [Chang et al., 2020] to ensure the features' robustness for downstream tasks. Here, we

formulate the reconstruction procedures as:

$$\mathbf{z}_w^r = g_r\left(\mathbf{z}_w^p \odot \mathbf{z}_w^c\right), \quad \mathbf{z}_w^{cr} = g_r\left(\mathbf{z}_{w'}^p \odot \mathbf{z}_w^c\right), \qquad (6)$$

where $g_r$ is the parameterized reconstruction model and $\odot$ is a free-to-choose fusion operators, such as element-wise product or concatenation. The reconstruction procedures are optimized by minimizing the entropy $H\left(\mathbf{z}_w \mid \mathbf{z}_{w'}^p, \mathbf{z}_w^c\right)$, where $w = w'$ or $w \neq w'$. Ideally, we reach the optimal sufficiency and disentanglement conditions illustrated in Figure 1(b) and 1(c) iff $H\left(\mathbf{z}_w \mid \mathbf{z}_{w'}^p, \mathbf{z}_w^c\right) = -\mathbb{E}_{p\left(\mathbf{z}_w, \mathbf{z}_{w'}^p, \mathbf{z}_w^c\right)}\left[\log p\left(\mathbf{z}_w \mid \mathbf{z}_{w'}^p, \mathbf{z}_w^c\right)\right] = 0$, where $\mathbf{z}_w$ is exactly recovered given its complementary representation and the predictive representation of any view. Nevertheless, the condition probability $p\left(\mathbf{z}_w \mid \mathbf{z}_{w'}^p, \mathbf{z}_w^c\right)$ is intractable, we hence use the variational distribution approximated by $g_r$ instead, denoted as $q\left(\mathbf{z}_w \mid \mathbf{z}_{w'}^p, \mathbf{z}_w^c\right)$. We provide the upper bound of $H\left(\mathbf{z}_w \mid \mathbf{z}_{w'}^p, \mathbf{z}_w^c\right)$ in Theorem 2.

**Theorem 2.** *Assume $q$ is a Gaussian distribution, $g_r$ is the parameterized reconstruction model which infers $\mathbf{z}_w$ from ($\mathbf{z}_{w'}^p, \mathbf{z}_w^c$). Then we have:*

$$H\left(\mathbf{z}_w \mid \mathbf{z}_{w'}^p, \mathbf{z}_w^c\right) \leqslant \left\| \mathbf{z}_w - g_r\left(\mathbf{z}_{w'}^p \odot \mathbf{z}_w^c\right) \right\|_2^2,$$

where $w = w'$ or $w \neq w'$. The detailed proof is demonstrated in Appendix E. Since we adopt two augmentation views, the objective function constraining representation disentanglement can be formulated as:

$$\mathcal{L}_{\text{recon}} = \frac{1}{2N} \sum_{i=1}^{N} \sum_{w=1}^{2} \left[ \left\| \mathbf{z}_{w,i} - \mathbf{z}_{w,i}^r \right\|_2^2 + \left\| \mathbf{z}_{w,i} - \mathbf{z}_{w,i}^{cr} \right\|_2^2 \right]. \quad (7)$$

### 3.2 ADVERSARIAL CONTRASTIVE VIEW

With the cross-view reconstruction mechanism above, the two learned representations stated above are optimized toward the disentangled manner. However, it is still necessary to further prevent the learned predictive representation from focusing on the partial features, because we do not have access to the explicit domain knowledge and such a small scope will increase the risk of a shortcut solution. Therefore, we extend the Equation 5 to three contrastive views and add an extra global view without topological perturbation as the third view to guarantee the learned $\mathbf{z}^p$ maintain the global semantics instead of partial or even trivial features, i.e., $\mathbf{z}_1^p \sim G$ and $\mathbf{z}_2^p \sim G$. During the experiments, we find an adversarial graph sample perturbed from the original graph view that can help the model achieve stronger robustness. A possible explanation is that there is still redundant information that is not predictive left in the shared information of the two $\mathbf{z}^p$'s in the two augmentation views, especially when the implemented augmentations are moderate. An adversarial view may further alleviate redundancy. We define the adversarial objective as follows:

$$\delta^* = \underset{\|\delta\|_\infty \leqslant \epsilon}{\arg\max} \mathcal{L}_{\text{adv}}\left(\mathbf{z}_1^p, \mathbf{z}_2^p, \mathbf{z}_{\text{adv}} + \delta\right), \qquad (8)$$

where the adversarial sample $\mathbf{z}_{\text{adv}} + \delta$ together with the two augmentation views, i.e., $\mathbf{z}_1^p$ and $\mathbf{z}_2^p$ are employed as the positive pair. Our crafted perturbation is spurred by recent work [Yang et al., 2021] that add perturbation $\delta$ on the output of first hidden layer $\mathbf{h}^{(1)}$, since it is empirically proved to generate more challenging views than adding perturbation on the initial node feature. Therefore, the adversarial contrastive objective is defined as:

$$\mathcal{L}_{\text{adv}} = \frac{1}{N} \sum_{i=1}^{N} \sum_{w=1}^{2} \max_{\delta*} \mathcal{L}_{\text{CL}} \left( \mathbf{z}_{w,i}^p, \mathbf{z}_{\text{adv}} + \delta^* \right), \quad (9)$$

where the optimized perturbation $\delta'$ is solved by projected gradient descent (PGD) [Madry et al., 2018]. Finally, we derive the joint objective of GCVR by combining all of the objectives above together. The joint objective is as follows:

$$\min_{f,g} \mathbb{E}_{G \in \mathbf{G}} \left[ \mathcal{L}_{\text{pre}} + \lambda_r \mathcal{L}_{\text{recon}} + \lambda_a \max_{\|\delta\|_\infty \leqslant \epsilon} \mathcal{L}_{\text{adv}} \right], \quad (10)$$

where $\lambda_r$ and $\lambda_a$ are the coefficients to balance the magnitude of each loss term. Our proposed model is able to approximate the optimal representation illustrated in Figure 1(c) with the joint objective.

# 4 EXPERIMENTS

In this section, we first demonstrate the empirical evaluation results of our proposed GCVR and the state-of-the-art baselines on multiple public graph benchmark datasets under different settings. An ablation study is also included to evaluate the effectiveness of the designs in GCVR. We further conduct experiments to study the robustness and the representation disentanglement of the proposed GCVR with extensive experiments. More content about dataset statistics, training details, and other empirical analyses are provided in Appendix C, D, F and G.

## 4.1 EXPERIMENTAL SETUPS

**Datasets.** For the unsupervised learning setting, we evaluate our model on five graph benchmark datasets from the field of bioinformatics, including MUTAG, PTC-MR, NCI1, DD, and PROTEINS, and the other four are from the field of social networks, which are COLLAB, IMDB-B, RDT-B, and IMDB-M, for the task of graph-level property classification. For the transfer learning setting, we follow previous work [You et al., 2020, Xu et al., 2021b] to pre-train our model on the ZINC-2M dataset, which contains 2 million unlabeled molecule graphs sampled from MoleculeNet [Wu et al., 2018a], then evaluate its performance on eight binary classification datasets from chemistry domain, where the eight datasets are split according to the scaffold to simulate the out-of-distribution scenario in real-world. Additionally, We use ogbg-molhiv from Open Graph Benchmark Dataset

[Hu et al., 2020a] to evaluate our model over large-scale datasets under the semi-supervised setting. We provide more details about dataset statistics in Appendix C.

**Baselines.** Under the unsupervised learning setting, we compare GCVR with the eight SOTA self-supervised learning methods, including GraphCL [You et al., 2020], Info-Graph[Sun et al., 2019], MVGRL [Hassani and Khasahmadi, 2020], AD-GCL[Suresh et al., 2021], GASSL[Yang et al., 2021], InfoGCL[Xu et al., 2021a], RGCL [Li et al., 2022] and DGCL[Li et al., 2021], as well as three classical unsupervised representation learning methods, including node2vec [Grover and Leskovec, 2016], graph2vec [Narayanan et al., 2017], and GVAE [Kipf and Welling, 2016]. For the transfer learning setting, we employ AttrMasking [Hu et al., 2020b], ContextPred [Hu et al., 2020b], GraphCL [You et al., 2020], GraphLoG [Xu et al., 2021b], AD-GCL [Suresh et al., 2021], RGCL [Li et al., 2022] and GraphMAE [Hou et al., 2022] as baselines to evaluate the effectiveness of our proposed GCVR. Besides, we also compare our proposed methods with GraphCL, SimGRACE Xia et al. [2022], AutoGCL Yin et al. [2022] and DCL as the baselines to evaluate the effectiveness of our proposed GCVR under the semi-supervised learning setting.

**Evaluation Protocol.** For the unsupervised setting, we follow the evaluation protocols of previous works [Sun et al., 2019, You et al., 2020] to verify the effectiveness of our model. The learned representation is fine-tuned by a linear SVM classifier for task-specific prediction. We report the mean test accuracy evaluated by 10-fold cross-validation with the standard deviation of five random seeds as the final performance. For the transfer learning setting, we follow the finetuning procedures of previous work [You et al., 2020] and report the mean ROC-AUC scores with a standard deviation of 10 repeated runs on each downstream dataset as the final performance. In addition, we follow the setting of semi-supervised representation learning from GraphCL on the ogbg-molhiv dataset, with the finetune label rates as 1%, 10%, and 20%. The final performance is reported as the mean ROC-AUC score of five repeated runs with different initialization random seeds.

## 4.2 OVERALL PERFORMANCE COMPARISON

**Unsupervised learning.** The overall performance comparison is shown in Table 1 and we can have three observations: (1) Graph Contrastive Learning (GCL)-based methods consistently outperform traditional unsupervised learning techniques, underscoring the benefits of incorporating instance-level supervision. (2) The models RGCL, AD-GCL, and GASSL exhibit advantages compared to GraphCL. This finding lends empirical support to the hypothesis that the InfoMax objective may introduce excessive redundant information, leading to issues with feature suppression. (3) Notably, our proposed models, GCVR and DGCL, consis-

Table 1: Overall comparison on multiple graph classification benchmarks under unsupervised learning setting. Results are reported as mean±std%, the best performance is bolded and runner-ups are underlined. "-" indicates the result is not reported in the original papers.

| | MUTAG | PTC-MR | COLLAB | NCI1 | PROTEINS | IMDB-B | RDT-B | IMDB-M | DD |
|---|---|---|---|---|---|---|---|---|---|
| node2vec | 72.6±10.2 | 58.6±8.0 | - | 54.9±1.6 | 57.5±3.6 | - | - | - | - |
| graph2vec | 83.2±9.3 | 60.2±6.9 | - | 73.2±1.8 | 73.3±2.1 | 71.1±0.5 | 75.8±1.0 | 50.4±0.9 | - |
| InfoGraph | 89.0±1.1 | 61.7±1.4 | 70.7±1.1 | 76.2±1.1 | 74.4±0.3 | 73.0±0.9 | 82.5±1.4 | 49.7±0.5 | 72.9±1.8 |
| VGAE | 87.7±0.7 | 61.2±1.8 | - | - | - | 70.7±0.7 | 87.1±0.1 | 49.3±0.4 | - |
| MVGRL | 89.7±1.1 | 62.5±1.7 | - | - | - | 74.2±0.7 | 84.5±0.6 | 51.2±0.5 | - |
| GraphCL | 86.8±1.3 | 63.6±1.8 | 71.4±1.2 | 77.9±0.4 | 74.4±0.5 | 71.1±0.4 | 89.5±0.8 | - | 78.6±0.4 |
| InfoGCL | 91.2±1.3 | 63.5±1.5 | 80.0±1.3 | 80.2±0.6 | - | 75.1±0.9 | - | 51.4±0.8 | - |
| DGCL | 92.1±0.8 | 65.8±1.5 | **81.2±0.3** | 81.9±0.2 | 76.4±0.5 | **75.9±0.7** | 91.8±0.2 | 51.9±0.4 | - |
| AD-GCL | 89.7±1.0 | - | 73.3±0.6 | 69.7±0.5 | 73.8±0.5 | 72.3±0.6 | 85.5±0.8 | 49.9±0.7 | 75.1±0.4 |
| RGCL | 87.7±1.0 | - | 70.9±0.7 | 78.1±1.1 | 75.0±0.4 | 71.9±0.8 | 90.3±0.6 | - | 78.9±0.5 |
| GASSL | 90.9±7.9 | 64.6±6.1 | 78.0±2.0 | 80.2±1.9 | - | 74.2±0.5 | - | 51.7±2.5 | - |
| GraphMAE | 91.2±1.3 | - | 80.3±0.5 | 80.4±0.3 | 75.3±0.4 | 75.5±0.7 | 88.0±0.2 | 51.6±0.5 | - |
| **GCVR** | **92.3±0.7** | **67.4±1.3** | 80.5±0.5 | **82.0±1.0** | **76.8±0.4** | 75.6±0.4 | **92.4±0.9** | **52.2±0.5** | **80.5±0.5** |

tently surpass other baseline models in performance, illustrating the efficacy of disentangled representation. Particularly, GCVR achieves state-of-the-art results on the majority of datasets, highlighting its effectiveness in this domain. We think the possible reason behind the impressive performances of DGCL on COLLAB and IMDB-B could stem from its adaptable setting of the disentanglement head number, which enables larger hyperparameter search space but also requires more effort to find the best configuration. Though less optimal on COLLAB and IMDB-B compared with DGCL, our proposed GCVR achieves the best performance on all the other datasets under an unsupervised learning setting.

**Transfer learning.** Table 2 presents the experimental outcomes in the context of transfer learning. In this setting, the 'No Pre-Train' approach omits the self-supervised pre-training phase on the ZINC-2M dataset before the fine-tuning process. The results presented in Table 2 illustrate that no baseline achieves consistently superior performance across all eight datasets, which includes advanced models like GraphLoG and GraphMAE. On the other hand, several competitive baselines, such as AttrMasking and ContextPred, benefit from the incorporation of domain-specific knowledge during training Hu et al. [2020b], however, all the graph SSL baselines and our proposed GCVR are in the absence of such specialized knowledge. Under this condition, our proposed GCVR still achieves the best performance on three of the eight datasets and the second-best performance on another three datasets. Notably, GCVR achieves the highest average performance across the datasets. These results demonstrate GCVR's notable proficiency in addressing transfer learning challenges, affirming its effectiveness in this demanding context. In the meantime, JOAO, RGCL, and GCVR, all derivatives of GraphCL, surpass GraphCL in average performance. This finding empirically shows the

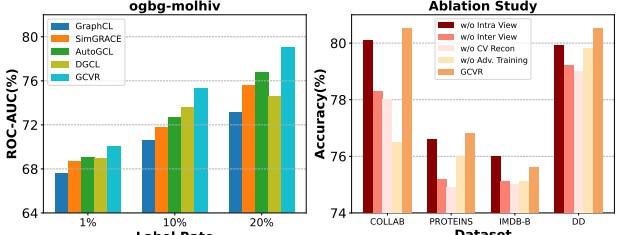

Figure 3: (right) Performance comparison of semi-supervised learning on ogbg-molhiv. (left) Performance comparison between the GCVR and four model variants.

detrimental impact of biased information in these models and highlights the critical need for strategies to mitigate such biases.

**Semi-supervised learning.** The experimental results, illustrated in Figure 3, show that our model significantly outperforms all baselines across three label-rate fine-tuning scenarios. Notably, there is a clear correlation between increasing label rates and performance improvements, with gains of 1.3%, 2.4%, and 2.9% observed at label rates of 1%, 10%, and 20%, respectively. This trend may be explained by the hypothesis that higher volumes of trainable data introduce more redundant information, thereby exacerbating the feature suppression problem. The effective removal of this redundant information is crucial, as it seems to play a key role in the observed enhancements in performance.

## 4.3 ABLATION STUDY

To further assess the individual contributions of the various modules in our proposed GCVR, we conducted ablation studies. These studies involved the construction of two mod-

Table 2: Overall comparison on multiple graph classification benchmarks under transfer learning setting. Results are reported as mean±std%, the best performance is bolded and runner-ups are underlined.

|  | BBBP | Tox21 | ToxCast | SIDER | ClinTox | MUV | HIV | BACE | AVG |
|---|---|---|---|---|---|---|---|---|---|
| No Pre-Train | 65.8±4.5 | 74.0±0.8 | 63.4 ±0.6 | 57.3±1.6 | 58.0±4.4 | 71.8±2.5 | 75.3±1.9 | 70.1±5.4 | 67.0 |
| AttrMasking | 64.3±2.8 | **76.7±0.4** | **64.2±0.5** | 61.0±0.7 | 71.8±4.1 | 74.7±1.4 | 77.2±1.1 | 79.3±1.6 | 71.1 |
| ContextPred | 68.0±2.0 | 75.7±0.7 | 63.9±0.6 | 60.9±0.6 | 65.9±3.8 | 75.8±1.7 | 77.3±1.0 | 79.6±1.2 | 70.9 |
| GraphCL | 69.5±0.5 | 75.4±0.9 | 63.8±0.4 | 60.8±0.7 | 70.1±1.9 | 74.5±1.3 | 77.6±0.9 | 78.2±1.2 | 70.8 |
| GraphLoG | **72.5±0.8** | 75.7±0.5 | 63.5±0.7 | 61.2±1.1 | 76.7±3.3 | 76.0±1.1 | 77.8±0.8 | **83.5±1.2** | 73.4 |
| JOAO | 70.2±1.0 | 75.0±0.3 | 62.9±0.5 | 60.0±0.8 | 81.3±2.5 | 71.7±1.4 | 76.7±1.2 | 51.5±0.4 | 71.9 |
| RGCL | 71.4±0.7 | 75.2±0.3 | 63.3±0.2 | 61.4±0.6 | 83.4 ±0.9 | **76.7 ±1.0** | 77.9±0.8 | 76.0±0.8 | 73.2 |
| GraphMAE | 72.0±0.6 | 75.5±0.6 | 64.1±0.3 | 60.3±1.1 | 82.3 ±1.2 | 76.3 ±2.4 | 77.2±1.0 | 83.1±0.9 | 73.8 |
| **GCVR** | 72.1±0.5 | 75.9±0.6 | 63.0±0.5 | **62.2±0.7** | **83.6±1.5** | 76.6±0.7 | **78.1±1.1** | 80.8±1.8 | **74.0** |

ified versions of the model: (1)**w/o Intra**, which excludes the intra-view reconstruction; (2)**w/o Inter**, which excludes the inter-view reconstruction; (3) **w/o CV Recon**, which completely excludes the cross-view reconstruction process; and (4) **w/o Adv. Training**, which omits the adversarial training component. The performance of these variants is demonstrated in the left subplot of Figure 3. An analysis of the results indicates that the integration of reconstruction from both views and adversarial view in the GCVR model yields superior performance compared to the variants. The absence of the reconstruction process from either view can impede the model's ability to optimize representations in a disentangled fashion, as evidenced in Figure 1(c). This omission leads to persistent issues with feature suppression in the resultant representations. Additionally, the model variant lacking the adversarial view exhibits tendencies towards representation collapse and accrues unnecessary redundant information, resulting in less optimal performance in downstream tasks.

## 4.4 ROBUSTNESS ANALYSIS

In this section, additional experiments are conducted on the ogbg-molhiv dataset to assess the robustness of representation under aggressive augmentation and perturbation. The corresponding results are presented in the left two subplots of Figure 4. Our method is compared with GASSL across varying perturbation bounds and attack steps to evaluate their resiliences against adversarial attacks. Given that both our model and GASSL utilize the GIN as the underlying backbone network, the performance of GIN is also included as a baseline for comparison. Despite the notable performance decline induced by aggressive adversarial attacks, our proposed GCVR model achieves comparable or better performance with GIN across a majority of perturbation scenarios and demonstrates more impressive resilience than that of GASSL.

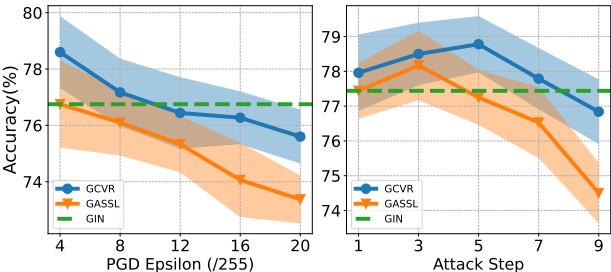

Figure 4: The model performance on ogbg-molhiv under different perturbation bound and attack steps.

## 4.5 DISENTANGLEMENT ANALYSIS

To investigate whether the feature suppression problem is equally serious in $\mathbf{z}^p$ and $\mathbf{z}^c$, we conduct experiments to compare the performance of the two representations on downstream tasks. The comparison results are shown in Table 3. It is easy to observe that there is a large performance gap between the two learned representations, indicating the different feature suppression issues between them and the features subset that are more robust to augmentation is more informative and transferable than those sensitive to augmentations. To further study the influence of the disentanglement design in GCVR on the optimization process, we use the InfoNCE loss [van den Oord et al., 2018] to dynamically measure the representation difference between the two augmentation graph views based on the two disentangled representations. For simplicity, we only demonstrate the first 100 pre-training epochs of PROTEINS and COLLAB in Figure 5, we can observe similar phenomena on other datasets. From the loss curves in Figure 5 we can find that contrastive loss between predictive representations, i.e., $\mathbf{z}^p$, gradually decreases, indicating the predictive representation is optimized to capture all the shared information between the two augmentation views. Meanwhile, the loss between the non-predictive representations, i.e., $\mathbf{z}^c$, achieves a noticeable increase, which is consistent with our expectation that

Table 3: Performance comparison of the two learned representations. Results are reported as mean±std%.

|  | MUTAG | PTC-MR | COLLAB | NCI1 | PROTEINS | IMDB-B | RDT-B | IMDB-M | DD | ogbg-molhiv |
|---|---|---|---|---|---|---|---|---|---|---|
| $\mathbf{z}^c$ | 88.1±1.2 | 58.6±2.0 | 75.1±0.7 | 72.2±2.0 | 73.5±0.8 | 71.8±0.9 | 89.4±1.0 | 47.8±0.9 | 75.8±0.6 | 69.70±2.8 |
| $\mathbf{z}^p$ | **92.3±0.7** | **67.4±1.3** | **80.5±0.5** | **82.0±1.0** | **76.8±0.4** | **75.6±0.4** | **92.5±0.9** | **52.2±0.5** | **80.5±0.5** | **75.36±1.4** |

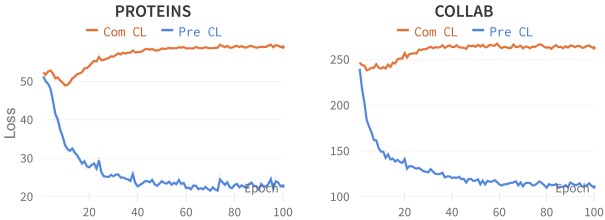

Figure 5: InfoNCE loss of the two disentangled representations, where orange lines are the InfoNCE loss between $\mathbf{z}_1^c$ and $\mathbf{z}_2^c$, blue lines are the InfoNCE loss between $\mathbf{z}_1^p$ and $\mathbf{z}_2^p$.

the two independent sampled augmentation operators cause a distribution shift between the two augmentation views. Given the empirical analysis above, we believe our proposed GCVR can further alleviate the feature suppression issue with the disentanglement design.

## 5 RELATED WORK

**Graph contrastive learning.** Contrastive learning was first proposed in the compute vision field [Chen et al., 2020] and has raised a surge of interest in the area of self-supervised graph representation learning. The principle behind contrastive learning is to utilize the instance-level identity as supervision and maximize the consistency between positive pairs in hidden space through the designed contrast mode. Previous graph contrastive learning works generally rely on various graph augmentation techniques [Veličković et al., 2019, Qiu et al., 2020, Hassani and Khasahmadi, 2020, You et al., 2020, Zhao et al., 2023] to generate positive pairs from original data as similar samples. Recent works in this field try to improve the effectiveness of graph contrastive learning by finding a more challenging view [Suresh et al., 2021, You et al., 2021] or adding adversarial perturbation [Yang et al., 2021]. However, most of the existing methods suffer from the feature suppression issue in contrastive learning [Chen et al., 2021, Robinson et al., 2021, Zhang et al., 2023], where the predictive features and trivial ones are equally possible to be omitted during the training phase. Our model is spared from the issue by proposing corresponding designs to discern the essential features from those trivial and easily disturbed ones.

**Disentangled representation learning.** Disentangled representation learning arises from the computer vision field [Hsieh et al., 2018, Zhao et al., 2021b] to disentangle the

heterogeneous latent factors of the representations, therefore making the representations more robust and interpretable [Bengio et al., 2013]. This idea has now been widely adopted in graph representation learning, [Liu et al., 2020, Ma et al., 2019] utilizes a neighborhood routing mechanism to identify the latent factors in the node representations. Some other generative models [Kipf and Welling, 2016, Simonovsky and Komodakis, 2018] utilize Variational Autoencoders to balance reconstruction and disentanglement. The study of learning disentangled representations also outspreads self-supervised graph learning [Li et al., 2021] by contrasting the factorized representations. Recent works [Wen et al., 2022] further demonstrate the impressive robustness and explainability of disentangled representations in dynamic graphs. Despite the significant benefit obtained from the representation disentanglement, the underlined excessive information could still overload the model, thus resulting in limited capacities. Our model targets the issue by removing the redundant information that is considered irrelevant to the graph property.

**Graph information bottleneck.** The Information bottleneck (IB) [Tishby et al., 2000] has been widely adopted as a critical principle of representation learning. A representation containing minimal yet sufficient information is considered to be in compliance with the IB principle and many works [Alemi et al., 2017, Shwartz-Ziv and Tishby, 2017, Federici et al., 2020] have empirically and theoretically proved that representation agrees with the IB principle is both informative and robust. Recently, the IB principle is also borrowed to guide the representation learning of graph structure data. Current methods [Wu et al., 2020, Xu et al., 2021a, Suresh et al., 2021, Li et al., 2022] usually design different regularizations to learn compressed yet informative representations following the IB principle. We follow the information bottleneck to learn the expressive and robust representation in this work.

## 6 CONCLUSION

In this paper, we study the feature suppression problem in self-supervised graph representation learning. To avoid the predictive features being suppressed in learned representation, we propose a novel model, namely GCVR, which is designed following the information bottleneck principle. The cross-view reconstruction in GCVR can disentangle those more robust and transferable features from those trivial ones. Meanwhile, we also add an adversarial view as the

third view of contrastive learning to guarantee the global semantics and further enhance representation robustness. In addition, we theoretically analyze the working mechanism of our design and derive the objective based on the analysis. Extensive experiments on multiple graph benchmark datasets and different settings prove the ability of GCVR to learn robust and transferable graph representation. In the future, we can explore how to come up with a practical objective to further decrease the upper bound of the mutual information between the disentangled representations and try to utilize more efficient training strategies to make the proposed model more time-saving on large-scale graphs.

## ACKNOWLEDGEMENTS

This work was partially supported by the NSF under grants IIS-2321504, IIS-2334193, IIS-2340346, IIS-2203262, IIS-2217239, CNS-2203261, and CMMI-2146076. Any opinions, findings, and conclusions or recommendations expressed in this material are those of the authors and do not necessarily reflect the views of the sponsors.

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

# Graph Contrastive Learning with Cross-View Reconstruction
## (Supplementary Material)

**Qianlong Wen**[1]    **Zhongyu Ouyang**[1]    **Chunhui Zhang**[2]    **Yiyue Qian**[1]    **Chuxu Zhang**[3]    **Yanfang Ye**[1]

[1]University of Notre Dame, Notre Dame, IN, USA
[2]Dartmouth College, Hanover, NH, USA
[3]Brandeis University, Waltham, MA, USA

## A    TRAINING ALGORITHM

In this section, we summarized the details of our proposed method in the following Algorithm.

---
**Algorithm 1** The training algorithm of **GCVR**

---
**Input:** Graph dataset $\mathcal{G} = \{G_i = (V_i, E_i)\}_{i=1}^N$; augmentation family $\mathcal{T}$; loss coefficient $\lambda_r$, $\lambda_a$; ascrent step $T$; ascent step size $\alpha$; perturbation bound $\epsilon$.
**Output:** The disentangled predictive representations $\mathbf{Z}^p = \{\mathbf{z}_i^p\}_{i=1}^N$
**for** each training epoch **do**
    **for** sampled minibatch $\mathcal{B} = \{G_i\}_{i=1}^{|\mathcal{B}|}$ **do**
        **for** $G_i \in \mathcal{B}$ **do**
            $\mathbf{z}_{1,i} = f\left(t_1(G_i)\right), \mathbf{z}_{2,i} = f\left(t_2(G_i)\right)$, where $t_1(\cdot), t_2(\cdot) \in \mathcal{T}$
            $\mathbf{z}_{1,i}^p = g_p\left(\mathbf{z}_{1,i}\right), \mathbf{z}_{2,i}^p = g_p\left(\mathbf{z}_{1,i}\right)$
            $\mathbf{z}_{1,i}^c = g_c\left(\mathbf{z}_{1,i}\right), \mathbf{z}_{2,i}^c = g_c\left(\mathbf{z}_{1,i}\right)$
        **end for**
        Calculate $\mathcal{L}_{\text{pre}}$ according to Equation 6
        Calculate $\mathcal{L}_{\text{recon}}$ according to Equation 8
        $\mathcal{L} \leftarrow \mathcal{L}_{\text{pre}} + \lambda_r \mathcal{L}_{\text{recon}}$
        $\delta_0 \leftarrow U(-\epsilon, \epsilon)$
    **end for**
    **for** each $t = 1$ to $T$ **do**
        Calculate the $\mathcal{L}_{\text{adv}}$ according to Equation 10
        $\delta_t \leftarrow \delta_{t-1} + \alpha \nabla_\delta \mathcal{L}_{\text{adv}}$      Update perturbation to maximize $\mathcal{L}_{\text{adv}}$
        $\mathcal{L} \leftarrow \mathcal{L} + \frac{\lambda_a}{T} \mathcal{L}_{\text{adv}}$
    **end for**
    Update the parameter $\theta$ of $f$ and $g$ with the gradient $\nabla_\theta \mathcal{L}(\theta, \mathcal{B})$ over a minibatch;
**end for**
$\mathbf{Z}^p = \{\mathbf{z}_i^p\}_{i=1}^N$, where $\mathbf{z}_i^p = g_p\left(f(G_i)\right)$

---

## B    OUT-OF-DISTRIBUTION SCENARIO ON GRAPH

In this section, we will illustrate the out-of-distribution scenario in the graph learning task. During molecule property study, A specific kind of property (e.g., toxicity and lipophilicity) of a molecule is usually dependent on if it has corresponding sub-structures (termed as functional group). For example, hydrophilic molecules usually have the oxhydryl group $(-OH)$ Therefore, a well-trained GNN model on molecule graph prediction task is capable of reflecting the sub-structure information in the graph representation. However, it is usually the case in a real-world scenario that the predictive functional group is usually accompanied by some irrelevant groups in some environments, thus causing spurious correlations. This correlation

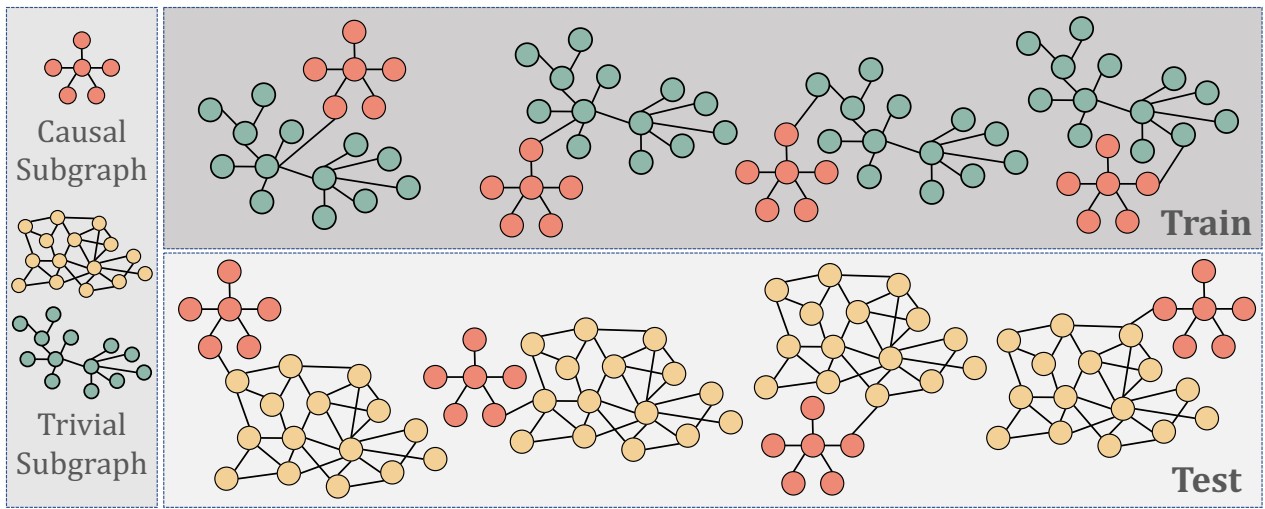

Figure 6: An out-of-distribution situation in molecule graph prediction task. The casual functional sub-structure (red) are spuriously correlated with different trivial sub-structures in training and test set. The statistical correlations can lead to poor robustness and transferability.

Table 4: Statistics of TU-datasets and OGB dataset.

| Dataset | #Graphs | Avg #Nodes | Avg #Edges | #Class | Metric | Category |
|---|---|---|---|---|---|---|
| MUTAG | 188 | 17.93 | 19.79 | 2 | Accuracy | biochemical |
| PTC-MR | 344 | 14.29 | 14.69 | 2 | Accuracy | biochemical |
| PROTEINS | 1,113 | 39.06 | 72.82 | 2 | Accuracy | biochemical |
| NCI1 | 4,110 | 29.87 | 32.30 | 2 | Accuracy | biochemical |
| DD | 1,178 | 284.32 | 715.66 | 2 | Accuracy | biochemical |
| COLLAB | 5,000 | 74.49 | 2457.78 | 3 | Accuracy | social network |
| IMDB-B | 1,000 | 19.77 | 96.53 | 2 | Accuracy | social network |
| RDT-B | 2,000 | 429.63 | 497.75 | 2 | Accuracy | social network |
| IMDB-M | 1,500 | 13.00 | 65.94 | 3 | Accuracy | social network |
| ogbg-molhiv | 41,127 | 25.50 | 27.50 | 2 | ROC-AUC | MoleculeNet |

usually leads to poor generalization performance when the model is evaluated in another environment with different spurious correlations. Figure 6 intuitively demonstrates this kind of scenario, where the red subgraph is the feature we can rely on to make the casual prediction. But it usually shows up with a green subgraph that does not serve as the functional graph of the property in the training set. Consequently, the model is easily misguided that the green subgraph is an important indicator of the property. When we evaluate the model on the testing set where the casual graph is correlated with another kind of group (yellow subgraph), there usually exists a huge gap between its performances on the two sets.

## C  SUMMARY OF DATASETS

In this work, we use nine datasets from TU Benchmark Datasets [Morris et al., 2020] to evaluate our proposed GCVR under the unsupervised setting, where five of them are biochemical datasets and the other four belong to social network datasets. We also utilize the ogng-molhiv dataset from Open Graph Benchmark (OGB) [Hu et al., 2020a] to further evaluate GCVR under the semi-supervised setting. Besides, the datasets sampled from MoleculeNet [Wu et al., 2018a] are employed to evaluate our model under the transfer learning setting. The statistics of these datasets are shown in Table 4 and 5.

All of the eleven datasets are publicly available, we attach their links as follows:

- TU datasets: `https://chrsmrrs.github.io/datasets/docs/datasets/`

- MoleculeNet datasets: `http://snap.stanford.edu/gnn-pretrain/`

- ogbg-molhiv dataset: `https://ogb.stanford.edu/docs/graphprop/#ogbg-mol`

Table 5: Statistics of MoleculeNet datasets.

| Dataset | #Graphs | Avg #Nodes | Avg Degree | #Tasks | Metric | Category |
|---------|---------|------------|------------|--------|--------|----------|
| ZINC-2M | 2,000,000 | 26.62 | 57.72 | - | - | biochemical |
| BBBP | 2,039 | 24.06 | 51.90 | 1 | ROC-AUC | biochemical |
| Tox21 | 7,813 | 18.57 | 38.58 | 12 | ROC-AUC | biochemical |
| ToxCast | 8,576 | 18.78 | 38.62 | 617 | ROC-AUC | biochemical |
| SIDER | 1,427 | 33.64 | 70.71 | 27 | ROC-AUC | biochemical |
| ClinTox | 1,477 | 26.15 | 55.76 | 2 | ROC-AUC | biochemical |
| MUV | 93,087 | 24.23 | 52.55 | 17 | ROC-AUC | biochemical |
| HIV | 41,127 | 25.51 | 54.93 | 1 | ROC-AUC | biochemical |
| BACE | 1,513 | 34.08 | 73.71 | 1 | ROC-AUC | biochemical |

## D    IMPLEMENTATION DETAILS

All experiments are conducted with the following settings:

- Operating System: Ubuntu 18.04.5 LTS

- CPU: AMD(R) Ryzen 9 3900x

- GPU: NVIDIA GeForce RTX 2080ti

- Software: Python 3.8.5; Pytorch 1.10.1; PyTorch Geometric 2.0.4; PyGCL 0.1.2; Numpy 1.20.1; scikit-learn 0.24.1.

We implement our framework with PyTorch and PyGCL library [Zhu et al., 2021]. We choose GIN [Xu et al., 2019] as the backbone graph encoder and the model is optimized through Adam optimizer. We follow [You et al., 2020, Yang et al., 2021, Li et al., 2021] to employ a linear SVM classifier for downstream task-specific classification. The graph augmentation operations used in our work are the same as [You et al., 2020], including node dropping, edge perturbation, attribute masking, and subgraph sampling, all of them are borrowed from the implementation of [Zhu et al., 2021]. There are two specific hyper-parameters in our model, namely $\lambda_r$ and $\lambda_a$, the search space of them are $\{0.0, 1.0, 3.0, 5.0, 10.0\}$ and $\{0.0, 0.25, 0.5, 0.75, 1.0\}$, respectively. For other important hyper-parameters, we find the best value of learning rate from $\{0.01, 0.005, 0.001, 0.0005, 0.0001\}$, embedding dimension from $\{32, 64, 128, 256, 512\}$, number of GNN layers from $\{2, 3, 4, 5\}$, batch size from $\{32, 64, 128, 256, 512\}$ (except for ogbg-molhiv $\{64, 128, 256, 512, 1024\}$). Besides, we fix the perturbation bound $\epsilon$, ascent step size $\alpha$, and ascent step $T$ as 0.008, 0.008, and 5 during hyper-parameter fine-tuning. For the implementation details of transfer learning, we follow the pre-training setting of previous works [You et al., 2020].

## E    PROOF

### E.1    PROOF OF THEOREM 1

**Theorem 1.** *Suppose $f(\cdot)$ is a GNN encoder as powerful as 1-WL test. Let $g_p(\cdot)$ elicits only the augmentation information from $\mathbf{z}$ meanwhile $g_c(\cdot)$ extracts the essential factors of $G$ from $\mathbf{z}_1$ and $\mathbf{z}_2$. Then we have:*

$$I\left(t_1(G); \mathbf{z}_2^c, \mathbf{z}_2^p\right) \geqslant I\left(\mathbf{z}_1^p; \mathbf{z}_2^p\right) \text{ where } G \in \mathcal{G} \text{ and } t_1(\cdot), t_2(\cdot) \in \mathcal{T}.$$

**Proof.** According to the assumption in Theorem 1, for any two graphs $G, G' \in \mathcal{G}$, if $G \cong G'$ then we have $\mathbf{z} = \mathbf{z}'$, where $\mathbf{z} = f(G)$ and $\mathbf{z}' = f(G')$.

Besides, $\mathbf{z}^p = g_p(\mathbf{z})$ is specific to the predictive factors and $\mathbf{z}^c = g_c(\mathbf{z})$ is particular to the non-predictive factors, which means $\mathbf{z}^p$ and $\mathbf{z}^c$ are mutually excluded and $\mathbf{z}^p \sim G$. So we have,

$$\begin{aligned} p\left(\mathbf{z}^p, \mathbf{z}^c\right) &= p\left(\mathbf{z}^p\right) p\left(\mathbf{z}^c\right) \\ p\left(\mathbf{z}^p, \mathbf{z}^c \mid t(G)\right) &= p\left(\mathbf{z}^p \mid t(G)\right) p\left(\mathbf{z}^c \mid t(G)\right). \end{aligned} \tag{11}$$

Then, we want to prove that given three random variables $a$, $b$ and $c$, if they satisfy $p(b, c) = p(b) p(c)$ and $p(b, c \mid a) =$

$p\left(b \mid a\right) p\left(c \mid a\right)$, we have $I\left(a, b \mid c\right) = I\left(a, b\right)$. According to the definition of mutual information, we have,

$$
\begin{aligned}
I\left(a; b \mid c\right) &= \\
&\sum_a \sum_b \sum_c p\left(a, b, c\right) \log \frac{p\left(a, b, c\right) p\left(c\right)}{p\left(a, c\right) p\left(b, c\right)} \\
&= \sum_a \sum_b \sum_c p\left(a\right) p\left(b, c \mid a\right) \log \frac{p\left(b, c \mid a\right) p\left(c\right)}{p\left(c \mid a\right) p\left(b\right) p\left(c\right)} \\
&= \sum_a \sum_b \sum_c p\left(a\right) p\left(b \mid a\right) p\left(c \mid a\right) \log \frac{p\left(b \mid a\right) p\left(c \mid a\right)}{p\left(c \mid a\right) p\left(b\right)} \\
&= \sum_a \sum_b p\left(a\right) p\left(b \mid a\right) \log \frac{p\left(b \mid a\right)}{p\left(b\right)} \\
&= \sum_a \sum_b p\left(a, b\right) \log \frac{p\left(b \mid a\right)}{p\left(b\right)} \\
&= I\left(a; b\right).
\end{aligned}
\tag{12}
$$

After that, by applying the chain rule to $I\left(t_1(G); \mathbf{z}_2^p, \mathbf{z}_2^c\right)$, we have,

$$
\begin{aligned}
I\left(t_1(G); \mathbf{z}_2^p, \mathbf{z}_2^c\right) &= I\left(t_1(G); \mathbf{z}_2^p \mid \mathbf{z}_2^c\right) + I\left(t_1(G); \mathbf{z}_2^c\right) \\
&\stackrel{(2)}{=} I\left(t_1(G); \mathbf{z}_2^p\right) + I\left(t_1(G); \mathbf{z}_2^c\right) \\
&\stackrel{(a)}{\geqslant} I\left(t_1(G); \mathbf{z}_2^p\right) \\
&\stackrel{(b)}{\geqslant} I\left(\mathbf{z}_1^c, \mathbf{z}_1^p; \mathbf{z}_2^p\right) \\
&\stackrel{(2)}{=} I\left(\mathbf{z}_1^c; \mathbf{z}_2^p\right) + I\left(\mathbf{z}_1^p; \mathbf{z}_2^p\right) \\
&\stackrel{(a)}{\geqslant} I\left(\mathbf{z}_1^p; \mathbf{z}_2^p\right),
\end{aligned}
\tag{13}
$$

where $\stackrel{(2)}{=}$ is derived from the conclusion we get in Equation 12, $\stackrel{(a)}{\geqslant}$ is based on the non-negativity of mutual information, i.e., $I(;) \geqslant 0$, and $\stackrel{(b)}{\geqslant}$ is because data processing inequality [Cover, 1999]. Finally, we reach to the lower bound of $I\left(t_1(G); \mathbf{z}_2^p, \mathbf{z}_2^c\right)$ in Equation 12, thus we can maximize the consistency between the information we capture from the two augmentation graph views by minimizing $\mathcal{L}_{\text{pre}}$.

## E.2   PROOF OF THEOREM 2

**Theorem 2.** *Assume $q$ is a Gaussian distribution, $g_r$ is the parameterized reconstruction model which infer $\mathbf{z}_w$ from $\left(\mathbf{z}_{w'}^p, \mathbf{z}_w^c\right)$. Then we have:*

$$
H\left(\mathbf{z}_w \mid \mathbf{z}_{w'}^p, \mathbf{z}_w^c\right) \leqslant \left\| \mathbf{z}_w - g_r\left(\mathbf{z}_{w'}^p \odot \mathbf{z}_w^c\right) \right\|_2^2 \text{ where } w = w' \text{ or } w \neq w'.
$$

**Proof.** To reconstruct the entangled representation $\mathbf{z}_w$ from its corresponding non-predictive representation $\mathbf{z}_w^c$ and the predictive representation of any augmentation view $\mathbf{z}_{w'}^p$ ($w$ and $w'$ are not necessarily equal), we need to minimize the conditional entropy:

$$
H\left(\mathbf{z}_w \mid \mathbf{z}_{w'}^p, \mathbf{z}_w^c\right) = -\mathbb{E}_{p\left(\mathbf{z}_w, \mathbf{z}_{w'}^p, \mathbf{z}_w^c\right)}\left[\log p\left(\mathbf{z}_w \mid \mathbf{z}_{w'}^p, \mathbf{z}_w^c\right)\right].
\tag{14}
$$

Since the real distribution of $p\left(\mathbf{z}_w \mid \mathbf{z}_{w'}^p, \mathbf{z}_{w'}^c\right)$ is unknown and intractable, we hereby introduce a variational distribution $q\left(\mathbf{z}_w \mid \mathbf{z}_{w'}^p, \mathbf{z}_w^c\right)$ to approximate it. Therefore, we have,

$$
\begin{aligned}
\mathbb{E}_{p\left(\mathbf{z}_w, \mathbf{z}_{w'}^p, \mathbf{z}_w^c\right)}\left[\log p\left(\mathbf{z}_w \mid \mathbf{z}_{w'}^p, \mathbf{z}_w^c\right)\right] &= \\
&\mathbb{E}_{p\left(\mathbf{z}_w, \mathbf{z}_{w'}^p, \mathbf{z}_w^c\right)}\left[\log q\left(\mathbf{z}_w \mid \mathbf{z}_{w'}^p, \mathbf{z}_w^c\right)\right] \\
&+ D_{\text{KL}}\left(p\left(\mathbf{z}_w \mid \mathbf{z}_{w'}^p, \mathbf{z}_w^c\right) \| q\left(\mathbf{z}_w \mid \mathbf{z}_{w'}^p, \mathbf{z}_w^c\right)\right).
\end{aligned}
\tag{15}
$$

Due to the non-negativity of KL-divergence between any two distributions, it is safe to say $-\mathbb{E}_{p\left(\mathbf{z}_w, \mathbf{z}_{w'}^p, \mathbf{z}_w^c\right)}\left[\log q\left(\mathbf{z}_w \mid \mathbf{z}_{w'}^p, \mathbf{z}_w^c\right)\right]$ is the upper bound of $H\left(\mathbf{z}_w \mid \mathbf{z}_{w'}^p, \mathbf{z}_w^c\right)$. Based on the assumption of Theorem 2, let $q\left(\mathbf{z}_w \mid \mathbf{z}_{w'}^p, \mathbf{z}_w^c\right)$ being a Gaussian distribution $\mathcal{N}\left(\mathbf{z}_w \mid g_r\left(\mathbf{z}_{w'}^p \odot \mathbf{z}_w^c\right), \sigma^2 \mathbf{I}\right)$, where $g_r(\cdot)$ is the reconstruct network that predict $\mathbf{z}_w$ from $(\mathbf{z}_{w'}^p, \mathbf{z}_w^c)$ and $\sigma$ is the variance. Thus we have,

$$
\begin{aligned}
H\left(\mathbf{z}_w \mid \mathbf{z}_{w'}^p, \mathbf{z}_w^c\right) &\leqslant -\mathbb{E}_{p\left(\mathbf{z}_w, \mathbf{z}_{w'}^p, \mathbf{z}_w^c\right)}\left[\log q\left(\mathbf{z}_w \mid \mathbf{z}_{w'}^p, \mathbf{z}_w^c\right)\right] \\
&= -\mathbb{E}_{p\left(\mathbf{z}_w, \mathbf{z}_{w'}^p, \mathbf{z}_w^c\right)}\left[\log\left(\frac{1}{\sqrt{2\pi I}\sigma} e^{-\frac{1}{2}\frac{\left(\mathbf{z}_w - g_r\left(\mathbf{z}_{w'}^p \odot \mathbf{z}_w^c\right)\right)^2}{(\sigma^2 \mathbf{I})}}\right)\right] \\
&= -\mathbb{E}_{p\left(\mathbf{z}_w, \mathbf{z}_{w'}^p, \mathbf{z}_w^c\right)}\left[\log\left(\frac{1}{\sqrt{2\pi I}\sigma}\right) - \frac{\left(\mathbf{z}_w - g_r\left(\mathbf{z}_{w'}^p \odot \mathbf{z}_w^c\right)\right)^2}{2\sigma^2 \mathbf{I}}\right].
\end{aligned}
\tag{16}
$$

Hence, we get the upper bound of $H\left(\mathbf{z}_w \mid \mathbf{z}_{w'}^p, \mathbf{z}_w^c\right)$ as Equation 16. To minimize the value of the unsolvable entropy, we can instead minimize the value of its upper bound and thereby derive the objective function as follow by neglecting the constant terms,

$$
\min \mathbb{E}_{p\left(\mathbf{z}_w, \mathbf{z}_{w'}^p, \mathbf{z}_w^c\right)} \left\|\mathbf{z}_w - g_r\left(\mathbf{z}_{w'}^p \odot \mathbf{z}_w^c\right)\right\|_2^2.
\tag{17}
$$

Since we adopt two augmentation views and propose the cross-view reconstruction mechanism in our method, we can minimize the entropy by minimizing $\mathcal{L}_{\text{recon}}$ and thus guarantee the disentanglement of $\mathbf{z}^p$ and $\mathbf{z}^c$.

## F   IMPACTS OF RECONSTRUCTION LOSS

In this work, we also conduct experiments to compare the effectiveness of different loss computation mechanisms for reconstruction. Except for the mean square error (MSE) loss in Equation 7, we also include the scaled cosine error (SCE) loss used in GraphMAE in this experiment. Previous work, like Barz and Denzler [2020], has demonstrated that Mean Squared Error (MSE) loss is sensitive to data scale, meaning its effectiveness can vary significantly with the range of target values. In contrast, the Scaled Cosine Error (SCE) utilized in GraphMAE is scale-insensitive, making it particularly effective in applications where the direction or orientation of vectors is crucial. Consequently, MSE loss is more suitable for regression problems where magnitude really matters, while cosine similarity loss usually can handle classification tasks better since it mainly focuses on the angle between vectors. This suggests a potential opportunity for enhancing our model performance on classification tasks by substituting MSE loss with SCE loss. To further investigate it, we test their effectiveness on four OGB datasets Hu et al. [2020a], where two of them (ogbg-molbbp and ogbg-moltox21) are classification tasks and the other two (ogbg-molesol and ogbg-molfreesolv) are regression tasks. The experimental results are shown in Table 6, from which we can see the results above also support the conclusion drawn in previous works. We believe the choice between MSE loss and SCE loss depends on the specific requirements of your task and the inherent properties of the data you are working with.

Table 6: Imapcts of different reconstruction loss computation methods.

|  | ogbg-molbbbp | ogbg-molbbbp | ogbg-molesol | ogbg-molfreesolv |
|---|---|---|---|---|
| **GCVR-MSE** | 70.1±0.8 | 73.3±0.5 | **1.112±0.040** | **4.032±0.575** |
| **GCVR-SCE** | **70.8±1.2** | **74.2±0.5** | 1.225±0.076 | 4.520 ± 0.680 |

## G   HYPER-PARAMETER SENSITIVITY

In this section, we study the impacts of some important hyper-parameters in our method, including reconstruction loss coefficient $\lambda_r$, adversarial loss coefficient $\lambda_a$, embedding dimension $d$, batch size $|\mathcal{B}|$ and number of GNN layers $L$. Here, we select four datasets, i.e., MUTAG, PROTEINS, RDT-B, and COLLAB, to report for simplicity because the four datasets cover different domains and scales. We illustrate the impacts of these hyper-parameters in the figures below.

From the result demonstrated in Figure 7, we can see the optimal reconstruction loss coefficient $\lambda_r$ is different dependent on the specific dataset, but all the values in our experiment can enhance the performance compared with the non-reconstruction variant, i.e., $\lambda_r = 0$, indicating the effectiveness of our proposed cross-view reconstruction mechanism.

Figure 8 shows that we could further raise the model performance through adversarial training, which proves a robust representation with less redundant information usually achieve more performance gain compared with the brittle one.

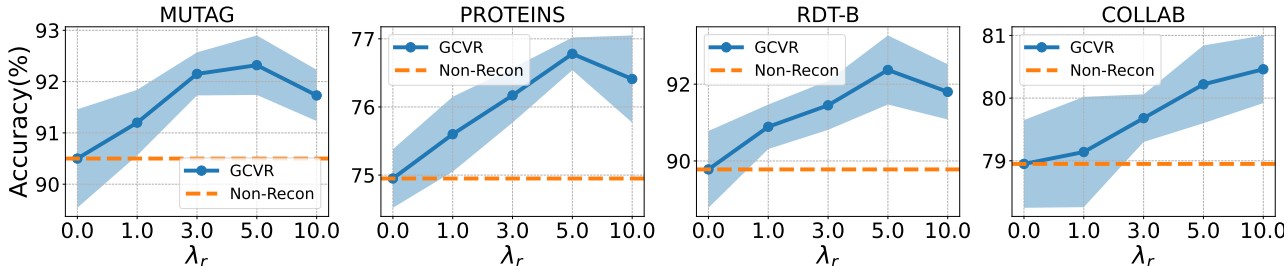

Figure 7: Impact of reconstruction loss coefficient $\lambda_r$ on different datasets, we specify the non-reconstruction situation ($\lambda_r = 0$) with the dashed line for comparison.

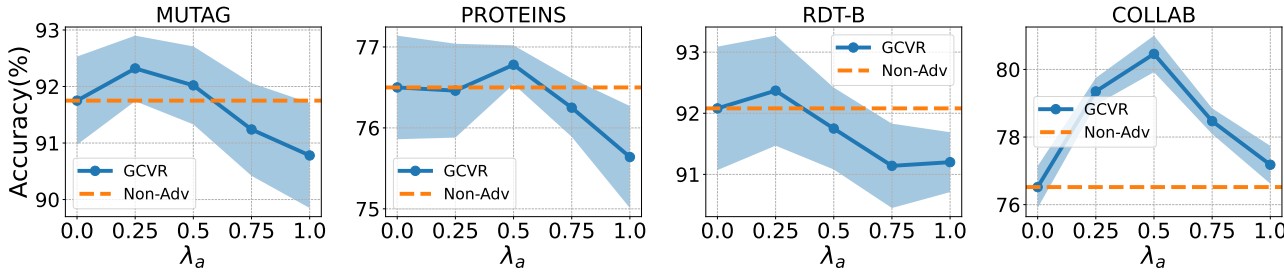

Figure 8: Impact of adversarial loss coefficient $\lambda_a$ xon different datasets, we specify the non-adversarial situation ($\lambda_a = 0$) with the dashed line for comparison.

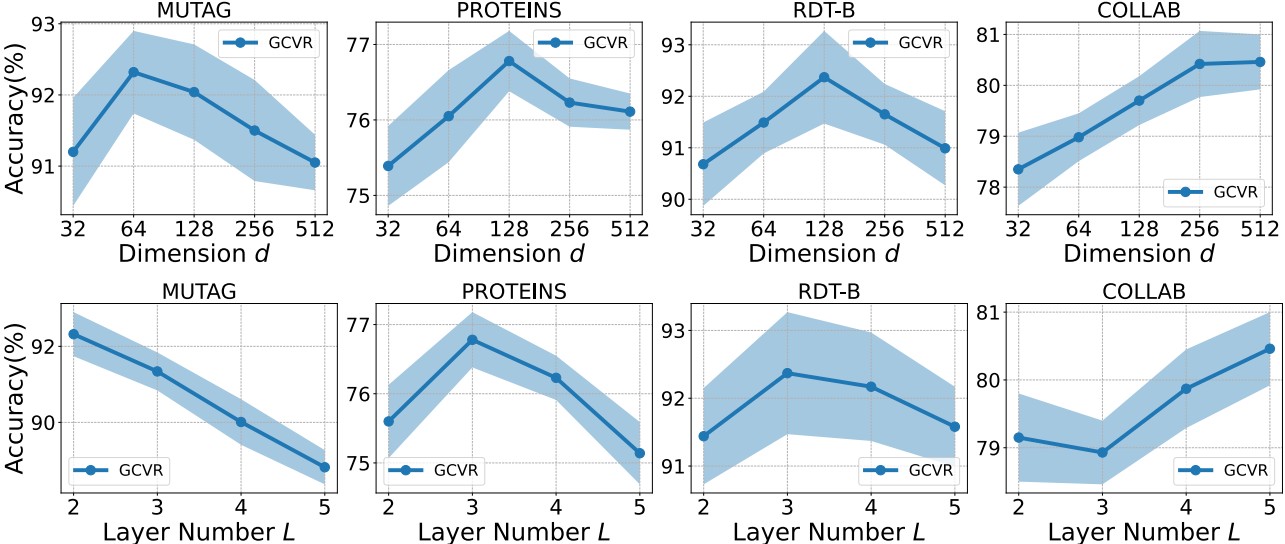

Figure 9: Impact of a embedding dimension $d$ and GNN layer number $L$ on different datasets.

During this process, we need to choose an appropriate adversarial loss coefficient $\lambda_a$, otherwise a too large $\lambda_a$ may hurt the information sufficiency of the learned representation. We put the impacts of embedding dimension $d$ and GNN layer number $L$ together because we can find a similar observation from their experimental results. From Figure 9, we observe that the optimal values of the two hyper-parameters generally increase as the dataset scale increases. The reason behind this phenomenon could be large datasets usually contain more latent factors than small datasets, therefore a model with a larger capacity is needed to fit the large datasets. However, such a high-capacity message-passing model will deteriorate the performance of a small dataset because it may cause the learned representation to over-smoothing and hence less informative.