# OpenReview forum: "GCVR: Reconstruction from Cross-View Enable Sufficient and Robust Graph Contrastive Learning"
_auai.org/UAI/2024/Conference — UAI 2024 poster_

### Official Review · Reviewer_9kmu · 2024-03-10

**Q2-1 Originality-Novelty:** 3
**Q2-2 Correctness-Technical Quality:** 3
**Q2-5 Clarity Of Writing:** 3

**Q1 Summary And Contributions:**

This paper explores a self-supervised learning strategy for graph data. It mainly introduces a cross-view reconstruction mechanism based on conventional graph contrastive learning. The proposed method shows better performance than previous methods as demonstrated in the experiments.

**Q2-3 Extent To Which Claims Are Supported By Evidence:**

3: Good: the main claims are supported by convincing evidence (in the form of adequate experimental evaluation, proofs, (pseudo-)code, references, assumptions).

**Q2-4 Reproducibility:**

3: Good: key resources (e.g. proofs, code, data) are available and key details (e.g. proofs, experimental setup) are sufficiently well-described for competent researchers to confidently reproduce the main results.

**Q3 Main Strengths:**

The proposed method shows better performance than previous methods

The writing is good, making this paper easy to follow.

**Q4 Main Weakness:**

- Mapping the feature into two different spaces ($z_c$ and $z_p$) is very similar to feature disentanglement. The authors should have a discussion.

- The authors should explore more for the complementary non-predictive factor and predictive information. What do they indicate? More analysis (quantitative and/or qualitative) is required.

**Q5 Detailed Comments To The Authors:**

See Q4.

**Q9 Complying With Reviewing Instructions:**

Yes

---

> ### Author Rebuttal · Authors · 2024-04-07
>
> # Response to Reviewer 9kmu
>
> Thank you for your valuable feedback and suggestions. We sincerely appreciate your acknowledgment of the contributions we make in this paper. Our detailed responses to your concerns are listed as follows.
>
> ### Q1. Mapping the feature into two different spaces ($\mathbf{z}^{p}$ and $\mathbf{z}^{c}$) is very similar to feature disentanglement. The authors should have a discussion.
>
> > Yes, as we introduced in our paper, the two representations are optimized to be disentangled with each other to avoid the predictive features being suppressed by the non-predictive features (introduced in the introduction and Appendix C). Unlike another disentanglement-based baseline in our study [1] that lacks regularization to enforce disentanglement, our designed cross-view reconstruction effectively separates the two types of feature sets, mitigating the issue of feature suppression. As a result, the extracted predictive representations are more informative and robust for downstream tasks, a finding that is empirically validated through the experiments presented in our research. We have a detailed discussion about the disentangled optimization goal we want to achieve in the introduction section and also discuss many previous disentanglement works in the related work section, please kindly refer to more details.
>
>
> ### Q2. The authors should explore more for the complementary non-predictive factor and predictive information. What do they indicate? More analysis (quantitative and/or qualitative) is required.
>
> > Thank you for raising this question. As described in our work, the non-predictive component ($\mathbf{z}^{c}$) refers to features that are susceptible to graph augmentation, meaning these features are inherently less robust, while the predictive represents the robust factors that are more essential to the predict tasks. Taking the molecular graph for illustration, the predictive components would be those important sub-structures (functional groups) that indicate the essential properties of the molecule, while the non-predictive components could be those trivial atoms and bonds.  In this work, we conduct experiments to compare the effectiveness of the two kinds of representations on the downstream prediction tasks, and the results are shown in Table 3 of our paper. It is easy to observe that there is a large performance gap between the two learned representations, indicating the features subset that are more robust to augmentation is more informative and transferable than those sensitive to augmentations. To further investigate their difference, we add another experiment to predict whether a graph is augmented or not with $\mathbf{z}^{c}$. Specifically, we will randomly apply augmentations on half of the graphs and leave the other half of graphs unchanged during the fine-tuning phases and train a linear classifier to predict whether the graph is augmented or not based on their generated $\mathbf{z}^{c}$. The results are as follows.
>
>
> |                  |      COLLAB     |     PROTEINS      |   IMDB-B      |  DD  |
> | :--------------: | :------------: | :------------: | :------------: | :------------: |
> | $\mathbf{z}^{c}$ |   96.5±2.2  |   92.4±3.1   |     94.0±1.6    |   97.8±1.0    |
> | $\mathbf{z}^{p}$ |   74.6±4.5  |   70.7±1.6   |     68.2±2.8    |   80.1±3.3     |
>
> > From the results we can see that $\mathbf{z}^{c}$ is more effective in predicting whether the graph is augmented or not than $\mathbf{z}^{p}$, showing their feature set difference within them.
>
> [1] Li et al. Disentangled Contrastive Learning on Graphs. In NeurIPS 2021
>
> **The content above is our response to your current review, please let us know if you have other questions and concerns and we are happy to respond.**

---

### Official Review · Reviewer_7NZD · 2024-03-21

**Q2-1 Originality-Novelty:** 2
**Q2-2 Correctness-Technical Quality:** 2
**Q2-5 Clarity Of Writing:** 3

**Q1 Summary And Contributions:**

The paper proposes a Graph Contrastive Learning with Cross View Reconstruction (GCVR) method to learn robust and sufficient representation from graph data. Specifically, GCVR introduces a cross-view reconstruction mechanism to elicit the essential features from raw graphs. Furthermore, GCVR introduces an extra adversarial view perturbed from the original view to pursue the intactness of the graph semantics and strengthen the representation robustness. GCVR outperforms other comparison methods on graph classification tasks on multiple benchmark datasets.

**Q2-3 Extent To Which Claims Are Supported By Evidence:**

2: Fair: the main claims are somewhat supported by evidence (but the experimental evaluation may be weak, or does not match entirely with the claims, important baselines may be missing, proofs contain important ideas but lack rigor, algorithmic details are only discussed superficially, references are imprecise, assumptions are not sufficiently motivated or explicated, etc.).

**Q2-4 Reproducibility:**

3: Good: key resources (e.g. proofs, code, data) are available and key details (e.g. proofs, experimental setup) are sufficiently well-described for competent researchers to confidently reproduce the main results.

**Q3 Main Strengths:**

The experimental results are promising.

**Q4 Main Weakness:**

See Detailed Comments To The Authors.

**Q5 Detailed Comments To The Authors:**

1. In my opinion, Eq. 7 and Eq. 8 are wrong.  $t_1(G)$ and $t_2(G)$ are vector representations after feature extraction by the GNN encoder, and $G$ represents a graph. Calculating CL loss between them is confusing to me. Furthermore, what the author expressed in Fig. 2 is $z_{adv}+\delta$, but it is another expression in Eqs. 7 and 8. There are inconsistencies in symbols in the paper. Therefore, the author should improve the writing of this paper.

2. Some classic graph contrastive learning methods [1], and [2] should also be compared. In addition, authors should also report experimental results in a semi-supervised setting.
[1] Xia J, Wu L, Chen J, et al. Simgrace: A simple framework for graph contrastive learning without data augmentation[C]//Proceedings of the ACM Web Conference 2022. 2022: 1070-1079.
[2] Yin Y, Wang Q, Huang S, et al. Autogcl: Automated graph contrastive learning via learnable view generators[C]//Proceedings of the AAAI conference on artificial intelligence. 2022, 36(8): 8892-8900.

3. The author used MSE loss for feature reconstruction in the reconstruction loss, but GraphMAE believes that the feature reconstruction effect of cosine loss is better. Can the author explain why this article uses MSE loss instead of cosine loss? I hope the authors compare the performance of MSE Loss and Cosine Loss in ablation experiments.

4. The performance of the method in this paper is very remarkable on the graph classification task, and I also want to know the effect of the method on the node classification task.

5. The author's results in the experimental setting of transfer learning are relatively poor, not as good as the reference [2]. Can the author explain the specific reasons?

6. In my opinion, the authors constructed a very large graph contrastive learning structure, i.e., a shared GNN encoder and five decoders. Therefore, the authors should compare the model complexity and running time of the model with other methods. Additionally, I would like to know if the improvement in model performance is due to the increase in the number of parameters.

7. For the augmented views $t_1(G)$ and $t_2(G)$, many details are not given, which confuses me. The author mentions in the supplementary material "The graph augmentation operations used in our work are the same as [You et al., 2020], including node dropping, edge perturbation, attribute masking, and subgraph sampling". The author used four graph augmentation operations, but the augmented views are only two. Furthermore, many details in the paper are not clearly described. I hope the author can open source the code of this paper during the rebuttal period to improve my understanding of this work.

8. The ablation experiment in this paper is very rough and only verifies the impact of adversarial training and reconstruction loss on the model. The author used inter-views and intra-views to reconstruct latent features but only verified the impact of inter-views on performance in the ablation experiment. In addition, the author used GIN for the GNN encoder part, but the author did not specify whether MLP or GIN was used for the decoder part. The authors should add experiments to illustrate the impact of different graph encoders and decoders (i.e., GCN, GAT, and GraphSAGE, etc.) on the experimental results.

9. The method proposed in this paper has two hyperparameter settings, i.e., $\lambda_r$ and $\lambda_a$. The author's experimental setup in the supplementary material section is unreasonable. The author sets one of the parameters to 0 to explore the experimental effect of the other parameter, which means that the adversarial training module or the reconstruction module does not work. The correct experimental setting should be to use a three-dimensional grid search to find the optimal $\lambda_r$ and $\lambda_a$.

**Q9 Complying With Reviewing Instructions:**

Yes

---

> ### Author Rebuttal · Authors · 2024-04-07
>
> # Response to Reviewer 7NZD (1/4)
>
> Thank you for your valuable feedback and suggestions. We sincerely appreciate your acknowledgment of the contributions we make in this paper. Our detailed responses to your concerns are listed as follows.
>
> ### Q1. In my opinion, Eq. 7 and Eq. 8 are wrong. $t_1(G)$ and $t_2(G)$ are vector representations after feature extraction by the GNN encoder, and $G$ represents a graph. Calculating CL loss between them is confusing to me. Furthermore, what the author expressed in Fig. 2 is $z_{adv}+\delta$, but it is another expression in Eqs. 7 and 8. There are inconsistencies in symbols in the paper. Therefore, the author should improve the writing of this paper.
>
> > Thank you for your suggestion, and we apologize for any confusion caused. We would like to provide further clarification on a particular point. As mentioned in the first paragraph of section 3.1, $t_1(\cdot)$ and $t_2(\cdot)$ represent two distinct augmentation operations, each independently sampled from the graph augmentation family. Therefore, $t_1(G)$ and $t_2(G)$ refer to augmented graphs instead of vector representations. Specifically, we use $\mathbf{z_1}$ and $\mathbf{z_2}$ to denote the vector representations obtained through the GNN encoder. Following this, two decoders are employed to isolate representations for predictive and non-predictive components, denoted by $\mathbf{z_1}^{p}$ ($\mathbf{z_2}^{p}$) and $\mathbf{z_1}^{c}$ ($\mathbf{z_2}^{c}$), respectively. The terms $t_1(G)$, $t_2(G)$, and $G$ are used in Equations (7) and (8) to maintain consistency with the notation in Figure 1. To eliminate any confusion in the revised manuscript, we will replace $t_1(G)$, $t_2(G)$, and $G+\delta$ with $\mathbf{z_1}^{p}$, $\mathbf{z_2}^{p}$, and $\mathbf{z_adv}$, respectively.
>
>
> ### Q2. Some classic graph contrastive learning methods [1], and [2] should also be compared. In addition, authors should also report experimental results in a semi-supervised setting.
>
> > Thanks for your suggestion. To further address your concern, we conduct experiments to evaluate the two methods [1,2] under the same semi-supervised setting in our paper.
>
>
> |                  |      1\%      |     10\%      |    20\%      |
> | :--------------: | :------------: | :------------: | :------------: |
> | SimGRACE |   68.7±0.4  |   71.8±0.5   |     75.6±0.6    |
> | AutoGCL |   69.1±0.7    |   72.7±0.6   |     76.8±0.8    |
> | GCVR |    70.0±0.6  |  75.4±0.5   |     79.0±0.8     |
>
> > From the results above we can see our GCVR can outperform the two baselines under the semi-supervised setting, which further proves the effectiveness of our proposed method in utilizing limited supervision.
>
>
>
> ### Q3. The author used MSE loss for feature reconstruction in the reconstruction loss, but GraphMAE believes that the feature reconstruction effect of cosine loss is better. Can the author explain why this article uses MSE loss instead of cosine loss? I hope the authors compare the performance of MSE Loss and Cosine Loss in ablation experiments.
>
> > Thanks for your insightful suggestion, we think there are both pros and cons of the two kinds of losses. Previous work, like [3], has demonstrated that Mean Squared Error (MSE) loss is sensitive to data scale, meaning its effectiveness can vary significantly with the range of target values. In contrast, the Scaled Cosine Error (SCE) utilized in GraphMAE is scale-insensitive, making it particularly effective in applications where the direction or orientation of vectors is crucial. Consequently, MSE loss is more suitable for regression problems where magnitude really matters, while cosine similarity loss usually can handle classification tasks better since it mainly focuses on the angle between vectors. This suggests a potential opportunity for enhancing our model performance on classification tasks by substituting MSE loss with SCE loss. To further investigate it, we conduct experiments to test their effectiveness on four OGB datasets [4], where two of them (ogbg-molbbp and ogbg-moltox21) are classification tasks and the other two (ogbg-molesol and ogbg-molfreesolv) are regression tasks. The experimental results are shown below.
>
> |          |      ogbg-molbbbp    |     ogbg-moltox21     |   ogbg-molesol     |  ogbg-molfreesolv  |
> | :--------------: | :------------: | :------------: | :------------: | :------------: |
> | GCVR-MSE |   70.1±0.8  |   73.3±0.5  |     **1.112±0.040**    |   **4.032 ± 0.575**    |
> | GCVR-SCE |   **70.8±1.2** |   **74.2±0.5**   |    1.225±0.076    |   4.520 ± 0.680    |
>
> > We can see the results above also support the conclusion drawn in previous works. We believe the choice between MSE loss and SCE loss depends on the specific requirements of your task and the inherent properties of the data you are working with.

---

### Official Review · Reviewer_g4ST · 2024-03-23

**Q2-1 Originality-Novelty:** 2
**Q2-2 Correctness-Technical Quality:** 3
**Q2-5 Clarity Of Writing:** 2

**Q1 Summary And Contributions:**

This paper proposes a graph contrastive learning with cross-view reconstruction (GCVR) framework for learning robust and sufficient graph embeddings. Theoretical analysis of the proposed method is provided and extensive experiments have been conducted to verify the effectiveness of the proposed method.

**Q2-3 Extent To Which Claims Are Supported By Evidence:**

2: Fair: the main claims are somewhat supported by evidence (but the experimental evaluation may be weak, or does not match entirely with the claims, important baselines may be missing, proofs contain important ideas but lack rigor, algorithmic details are only discussed superficially, references are imprecise, assumptions are not sufficiently motivated or explicated, etc.).

**Q2-4 Reproducibility:**

3: Good: key resources (e.g. proofs, code, data) are available and key details (e.g. proofs, experimental setup) are sufficiently well-described for competent researchers to confidently reproduce the main results.

**Q3 Main Strengths:**

S1. The paper is well organized and the theoretical proofs of the proposed method are provided.

S2. Experiments have been conducted by comparing with STOA graph contrastive learning methods and the results show the superiority of the proposed method.

**Q4 Main Weakness:**

W1. The motivation of the paper should be refined.

W2. Some experimental results should be discussed more clearly.

**Q5 Detailed Comments To The Authors:**

Q1. The presentation should be improved, e.g., the paragraphs in the introduction section are too long to read. Moreover, the authors claim that ‘GCL paradigm has been empirically and theoretically proved to be insufficient to learn robust and transferable representation’, but they also adopt GCL model in this paper, which is confusing.

Q2. DGCL performs better than GCVR on COLLAB and IMDB-B (Table 1), which should be discussed more clearly.

Q3. Some typos should be fixed, e.g., I(G^p; G^c) is not zero in Fig. 1 (c).

**Q9 Complying With Reviewing Instructions:**

Yes

---

> ### Author Rebuttal · Authors · 2024-04-07
>
> # Response to Reviewer g4ST
>
> Thank you for your valuable feedback and suggestions. We sincerely appreciate your acknowledgment of the contributions we make in this paper. Our detailed responses to your concerns are listed as follows.
>
> ### Q1. The presentation should be improved, e.g., the paragraphs in the introduction section are too long to read. Moreover, the authors claim that ‘GCL paradigm has been empirically and theoretically proved to be insufficient to learn robust and transferable representation’, but they also adopt GCL model in this paper, which is confusing.
>
> > Thanks for raising this question. As we illustrated in the introduction part, we acknowledge that the GCL paradigm has achieved impressive performances in various tasks. Meanwhile, it has been proved that there is also a potential issue during the optimization of the traditional GCL paradigm, i.e., how can we guarantee the learned representations are in accordance with the information bottleneck (IB). In this work, we are aiming to solve this issue and propose the GCVR to further improve the performance of GCL-based method, which is also demonstrated by the experiments in this paper.
> In the meantime, we will further polish the writing of the introduction part to make it shorter, all the modifications will be included in the final version of the paper as we are not allowed to update the paper manuscript at this time.
>
> ### Q2. DGCL performs better than GCVR on COLLAB and IMDB-B (Table 1), which should be discussed more clearly.
>
> > We think the possible reason behind the impressive performances of DGCL on COLLAB and IMDB-B could stem from its adaptable setting of the disentanglement head number, which enables larger hyperparameter search space but also requires more effort to find the best configuration. We would like to investigate GCVR and DGCL on more datasets and tasks for comparison, unfortunately, it seems the authors of DGCL did not release the source code. Though less optimal on COLLAB and IMDB-B compared with DGCL, our proposed GCVR achieves the best performance on all the other datasets under an unsupervised learning setting. Notably, GCVR significantly outperforms DGCL on the large-scale ogbg-molhiv dataset, where the out-of-distribution split setting amplifies the complexity. Therefore, we believe this evidence underpins our assertion that GCVR exhibits superior generalization capabilities across various datasets and demonstrates enhanced robustness compared to the baseline methods.
>
> ### Q3. Some typos should be fixed, e.g., I(G^p; G^c) is not zero in Fig. 1 (c).
>
> > To clarify, we use the solid green area to indicate the null set in Figure 1, as indicated in the figure's caption. Specifically, a graph $G$ could include two kinds of mutually excluded components, i.e., predictive components ($G^p$) and non-predictive components ($G^c$). Therefore, we denote the overlapping area between $G^p$ and $G^c$ with a null set (solid green). In practice, the learned graph representation $\mathbf{z}$ would include the information from both of $G^p$ and $G^c$ without additional regularization. In this work, we are aiming to separate the two parts from each other and store them in $\mathbf{z}^{p}$ and $\mathbf{z}^{c}$, respectively. Ideally, like the  the $G^p$ and $G^c$ in Figure 1(c), the learned $\mathbf{z}^{p}$ and $\mathbf{z}^{c}$ should be disentangled from each other, i.e., $I(\mathbf{z}^{p}; \mathbf{z}^{c})=0$.
>
>
> **The content above is our response to your current review, please let us know if you have other questions and concerns and we are happy to respond.**

---

### Official Review · Reviewer_vEvb · 2024-03-24

**Q2-1 Originality-Novelty:** 3
**Q2-2 Correctness-Technical Quality:** 3
**Q2-5 Clarity Of Writing:** 3

**Q1 Summary And Contributions:**

This study addresses the challenges of graph representation learning, particularly tackling the feature suppression problem in graph contrastive learning methods.

It introduces a new approach employing a cross-view reconstruction technique.

The effectiveness of this new method is substantiated through both theoretical analysis and empirical evaluations.

**Q2-3 Extent To Which Claims Are Supported By Evidence:**

3: Good: the main claims are supported by convincing evidence (in the form of adequate experimental evaluation, proofs, (pseudo-)code, references, assumptions).

**Q2-4 Reproducibility:**

3: Good: key resources (e.g. proofs, code, data) are available and key details (e.g. proofs, experimental setup) are sufficiently well-described for competent researchers to confidently reproduce the main results.

**Q3 Main Strengths:**

The concept is intuitive, and the presentation is clear.

The proposed method is well-supported by theoretical analysis, which aligns consistently with the design of the loss function.

Additionally, empirical results support the main claim.

**Q4 Main Weakness:**

The primary concerns are centered around two aspects:

1. Concerning the non-predictive component $\mathbf{z}^{c}$, suppose the model generates a noise term without considering the actual graph, denoted as $\mathbf{z}^{c} \sim p(\epsilon)$. Is there a mechanism or design within the model that penalizes such behavior? In other words, how can we verify, either theoretically or empirically, that $\mathbf{z}^{c}$ is indeed capturing non-predictive features from the data, rather than merely generating noise?
2. The second concern pertains to the experimental performance in a transfer learning context. It is evident that for certain datasets, such as ToxCast and BACE, the proposed method does not achieve the best results and is significantly outperformed by others. An explanation or analysis of these specific instances is required to understand the limitations of the proposed method.

**Q5 Detailed Comments To The Authors:**

See weakness

**Q9 Complying With Reviewing Instructions:**

Yes

---

> ### Author Rebuttal · Authors · 2024-04-07
>
> # Response to Reviewer vEvb
>
> Thank you for your valuable feedback and suggestions. We sincerely appreciate your acknowledgment of the contributions we make in this paper. Our detailed responses to your concerns are listed as follows.
>
>
> ### Concerning the non-predictive component $\mathbf{z}^c$, suppose the model generates a noise term without considering the actual graph, denoted as $\mathbf{z}^c \sim p(\epsilon)$. Is there a mechanism or design within the model that penalizes such behavior? In other words, how can we verify, either theoretically or empirically, that $\mathbf{z}^c$ is indeed capturing non-predictive features from the data, rather than merely generating noise?
>
> > Thank you for raising this question. We believe the cross-view reconstruction mechanism in our model can penalize such behavior. As described in our work, the non-predictive component refers to features that are susceptible to graph augmentation, meaning these features are inherently less robust. During training, we implement two distinct augmentation operations on a graph to produce two views, inducing a distribution shift between them. Through cross-view reconstruction, $\mathbf{z}^p$ is optimized to represent the shared feature set of the two views, exhibiting robustness to augmentation. Conversely, $\mathbf{z}^c$ is optimized to capture the complementary, less robust feature set, rather than random noise, to minimize reconstruction loss. Although $\mathbf{z}^c$ is less predictive compared to $\mathbf{z}^p$, we posit that it can still deliver superior prediction performance than random noise, as it is derived from actual graph data. This could explain why $\mathbf{z}_{c}$ outperforms random guessing, as evidenced in Table 3. To further investigate their difference, we add another experiment to predict whether a graph is augmented or not with $\mathbf{z}^{c}$. Specifically, we will randomly apply augmentations on half of the graphs and leave the other half of graphs unchanged during the fine-tuning phases and train a linear classifier to predict whether the graph is augmented or not based on their generated $\mathbf{z}^{c}$. The results are as follows.
>
>
> |                  |      COLLAB     |     PROTEINS      |   IMDB-B      |  DD  |
> | :--------------: | :------------: | :------------: | :------------: | :------------: |
> | $\mathbf{z}^{c}$ |   **86.5±2.2**  |   **82.4±3.1**   |     **79.0±1.6**    |   **90.8±1.0**    |
> | $\mathbf{z}^{p}$ |   71.6±4.5  |   70.7±1.6   |     62.2±2.8    |   71.1±3.3     |
>
> > From the results we can see that $\mathbf{z}^{c}$ is more effective in predicting whether the graph is augmented or not than $\mathbf{z}^{p}$, showing the feature set difference within them.
>
>
> ### The second concern pertains to the experimental performance in a transfer learning context. It is evident that for certain datasets, such as ToxCast and BACE, the proposed method does not achieve the best results and is significantly outperformed by others. An explanation or analysis of these specific instances is required to understand the limitations of the proposed method.
>
> > Thanks for bringing up this question. As we mentioned in the paper, transfer learning is a very challenging setting due to the distribution gap between the pre-training and fine-tuning datasets. The results presented in Table 2 illustrate that no baseline achieves consistently superior performance across all eight datasets, which includes advanced models like GraphLoG and GraphMAE. On the other hand, several competitive baselines, such as AttrMasking and ContextPred, benefit from the incorporation of domain-specific knowledge during training [1], however, all the graph SSL baselines and our proposed GCVR are in the absence of such specialized knowledge. Under this condition, our proposed GCVR still achieves the best performance on three of the eight datasets and the second-best performance on another three datasets. Notably, GCVR achieves the highest average performance across the datasets. These results demonstrate GCVR's notable proficiency in addressing transfer learning challenges, affirming its effectiveness in this demanding context.
>
>
> [1] Hu et al. Strategies for Pre-training Graph Neural Networks. In ICLR 2020
>
> **The content above is our response to your current review, please let us know if you have other questions and concerns and we are happy to respond.**

---

### Meta-Review · Area_Chair_ZSb6 · 2024-04-22

This paper proposes a graph contrastive learning method, which leverages cross-view reconstruction to learn graph representation. The proposed method shows better performance than previous methods as demonstrated in the experiments.

Most of the reviewers appreciate the theoretical analysis and promising results. The experimental results are also promising.  On the other hand, as the reviewers pointed out, the presentation can be further improved, and some aspects of the experimental results can be made clearer.